# ONLINE FACILITY LOCATION WITH PREDICTIONS

**Shaofeng H.-C. Jiang**
Peking University
Email: `shaofeng.jiang@pku.edu.cn`

**Erzhi Liu**
Shanghai Jiao Tong University
Email: `lezdzh@sjtu.edu.cn`

**You Lyu**
Shanghai Jiao Tong University
Email: `vergil@sjtu.edu.cn`

**Zhihao Gavin Tang**
Shanghai University of Finance and Economics
Email: `tang.zhihao@mail.shufe.edu.cn`

**Yubo Zhang**
Peking University
Email: `zhangyubo18@pku.edu.cn`

## ABSTRACT

We provide nearly optimal algorithms for online facility location (OFL) with predictions. In OFL, $n$ demand points arrive in order and the algorithm must irrevocably assign each demand point to an open facility upon its arrival. The objective is to minimize the total connection costs from demand points to assigned facilities plus the facility opening cost. We further assume the algorithm is additionally given for each demand point $x_i$ a natural prediction $f_{x_i}^{\mathrm{pred}}$, which is supposed to be the facility $f_{x_i}^{\mathrm{opt}}$ that serves $x_i$ in the offline optimal solution.

Our main result is an $O(\min\{\log \frac{n\eta_\infty}{\mathrm{OPT}}, \log n\})$-competitive algorithm where $\eta_\infty$ is the maximum prediction error (i.e., the distance between $f_{x_i}^{\mathrm{pred}}$ and $f_{x_i}^{\mathrm{opt}}$). Our algorithm overcomes the fundamental $\Omega(\frac{\log n}{\log \log n})$ lower bound of OFL (without predictions) when $\eta_\infty$ is small, and it still maintains $O(\log n)$ ratio even when $\eta_\infty$ is unbounded. Furthermore, our theoretical analysis is supported by empirical evaluations for the tradeoffs between $\eta_\infty$ and the competitive ratio on various real datasets of different types.

## 1 INTRODUCTION

We study the online facility location (OFL) problem with predictions. In OFL, given a metric space $(\mathcal{X}, d)$, a ground set $\mathcal{D} \subseteq \mathcal{X}$ of demand points, a ground set $\mathcal{F} \subseteq \mathcal{X}$ of facilities, and an *opening cost* function $w : \mathcal{F} \to \mathbb{R}_+$, the input is a sequence of demand points $X := (x_1, \ldots, x_n) \subseteq \mathcal{D}$, and the online algorithm must *irrevocably* assign $x_i$ to some *open* facility, denoted as $f_i$, upon its arrival, and an open facility cannot be closed. The goal is to minimize the following objective

$$\sum_{f \in F} w(f) + \sum_{x_i \in X} d(x_i, f_i),$$

where $F$ is the set of open facilities.

Facility location is a fundamental problem in both computer science and operations research, and it has been applied to various domains such as supply chain management (Melo et al., 2009), distribution system design (Klose & Drexl, 2005), healthcare (Ahmadi-Javid et al., 2017), and more applications can be found in surveys (Drezner & Hamacher, 2002; Laporte et al., 2019). Its online variant, OFL, was also extensively studied and well understood. The state of the art is an $O(\frac{\log n}{\log \log n})$-competitive deterministic algorithm proposed by Fotakis (2008), and the same work shows this ratio is tight.

We explore whether or not the presence of certain natural predictions could help to bypass the $O(\frac{\log n}{\log \log n})$ barrier. Specifically, we consider the setting where each demand point $x_i$ receives a

predicted facility $f_{x_i}^{\mathrm{pred}}$ that is supposed to be $x_i$'s assigned facility in the (offline) optimal solution. This prediction is very natural, and it could often be easily obtained in practice. For instance, if the dataset is generated from a latent distribution, then predictions may be obtained by analyzing past data. Moreover, predictions could also be obtained from external sources, such as expert advice, and additional features that define correlations among demand points and/or facilities. Previously, predictions of a similar flavor were also considered for other online problems, such as online caching (Rohatgi, 2020; Lykouris & Vassilvitskii, 2021), ski rental (Purohit et al., 2018; Gollapudi & Panigrahi, 2019) and online revenue maximization (Medina & Vassilvitskii, 2017).

## 1.1 OUR RESULTS

Our main result is a near-optimal online algorithm that offers a smooth tradeoff between the prediction error and the competitive ratio. In particular, our algorithm is $O(1)$-competitive when the predictions are perfectly accurate, i.e., $f_{x_i}^{\mathrm{pred}}$ is the nearest neighbor of $x_i$ in the optimal solution for every $1 \le i \le n$. On the other hand, even when the predictions are completely wrong, our algorithm still remains $O(\log n)$-competitive. As in the literature of online algorithms with predictions, our algorithm does not rely on the knowledge of the prediction error $\eta_\infty$.

**Theorem 1.1.** *There exists an $O(\min\{\log n, \max\{1, \log \frac{n\eta_\infty}{\mathrm{OPT}}\}\})$-competitive algorithm for the OFL with predictions, where $\eta_\infty := \max_{1 \le i \le n} d(f_{x_i}^{\mathrm{pred}}, f_{x_i}^{\mathrm{opt}})$ measures the maximum prediction error, and $f_{x_i}^{\mathrm{opt}}$ is the nearest neighbor of $x_i$ from the offline optimal solution* $\mathrm{OPT}$[1].

Indeed, we can also interpret our result under the robustness-consistency framework which is widely considered in the literature of online algorithms with predictions (cf. Lykouris & Vassilvitskii (2021)), where the robustness and consistency stand for the competitive ratios of the algorithm in the cases of arbitrarily bad predictions and perfect predictions, respectively. Under this language, our algorithm is $O(1)$-consistent and $O(\log n)$-robust. We also remark that our robustness bound nearly matches the lower bound for the OFL without predictions.

One might wonder whether or not it makes sense to consider a related error measure, $\eta_1 := \sum_{i=1}^{n} d(f_{x_i}^{\mathrm{pred}}, f_{x_i}^{\mathrm{opt}})$, which is the *total* error of predictions. Our lower bound result (Theorem 1.2) shows that a small $\eta_1$ is not helpful, and the dependence of $\eta_\infty$ in our upper bound is fundamental (see Section F for a more detailed explanation). The lower bound also asserts that our algorithm is nearly tight in the dependence of $\eta_\infty$.

**Theorem 1.2.** *Consider OFL with predictions with a uniform opening cost of $1$. For every $\eta_\infty \in (0, 1]$, there exists a class of inputs, such that no (randomized) online algorithm is $o(\frac{\log \frac{n\eta_\infty}{\mathrm{OPT}}}{\log \log n})$-competitive, even when $\eta_1 = O(1)$.*

As suggested by an anonymous reviewer, it might be possible to make the error measure $\eta_\infty$ more "robust" by taking "outliers" into account. A more detailed discussion on this can be found in Section G.

**Empirical evaluation.** We simulate our algorithm on both Euclidean and graph (shortest-path) datasets, and we measure the tradeoff between $\eta_\infty$ and the empirical competitive ratio, by generating random predictions whose $\eta_\infty$ is controlled to be around some target value. We compare with two baselines, the $O(\log n)$-competitive algorithm by Meyerson (2001) which do not use the prediction at all, and a naive algorithm that always trusts the prediction. Our algorithm significantly outperforms Meyerson's algorithm when $\eta_\infty \to 0$, and is still comparable to Meyerson's algorithm when $\eta_\infty$ is large where the follow-prediction baseline suffers a huge error.

We observe that our lead is even more significant on datasets with non-uniform opening cost, and this suggests that the prediction could be very useful in the non-uniform setting. This phenomenon seems to be counter-intuitive since the error guarantee $\eta_\infty$ only concerns the connection costs without taking the opening cost into consideration (i.e., it could be that a small $\eta_\infty$ is achieved by predictions of huge opening cost), which seems to mean the prediction may be less useful. Therefore, this actually demonstrates the superior capability of our algorithm in using the limited information from the predictions even in the non-uniform opening cost setting.

---

[1]When the context is clear, we also use $\mathrm{OPT}$ to denote the optimal solution.

Finally, we test our algorithm along with a very simple predictor that does not use any sophisticated ML techniques or specific features of the dataset, and it turns out that such a simple predictor already achieves a reasonable performance. This suggests that more carefully engineered predictors in practice is likely to perform even better, and this also justifies our assumption of the existence of a good predictor.

**Comparison to independent work.** A recent independent work by Fotakis et al. (2021a) considers OFL with predictions in a similar setting. We highlight the key differences as follows.

(i) We allow arbitrary opening cost $w(\cdot)$, while Fotakis et al. (2021a) only solves the special case of *uniform* facility location where all opening costs are equal. This is a significant difference, since as we mentioned (and will discuss in Section 1.2)), the non-uniform opening cost setting introduces outstanding technical challenges that require novel algorithmic ideas.

(ii) For the uniform facility location setting, our competitive ratio is $O(\log \frac{n\eta_\infty}{\text{OPT}}) = O(\log \frac{n\eta_\infty}{w})$, while the ratio in Fotakis et al. (2021a) is $\frac{\log(\text{err}_0)}{\log(\text{err}_1^{-1} \log(\text{err}_0))}$, where $\text{err}_0 = n\eta_\infty/w$, $\text{err}_1 = \eta_1/w$ and $w$ is the uniform opening cost. Our ratio is comparable to theirs when $\eta_1$ is relatively small. However, theirs seems to be better when $\eta_1$ is large, and in particular, it was claimed in Fotakis et al. (2021a) that the ratio becomes constant when $\text{err}_1 \approx \text{err}_0$. Unfortunately, we think this claim contradicts with our lower bound (Theorem 1.2) which essentially asserts that $\eta_1 = O(\eta_\infty)$ is not helpful for the ratio. Moreover, when $\eta_1$ is large, we find their ratio is actually not well-defined since the denominator $\log(\text{err}_1^{-1} \log(\text{err}_0))$ is negative. Therefore, we believe there might be some unstated technical assumptions, or there are technical issues remaining to be fixed, when $\eta_1$ is large.

Another independent work by Panconesi et al. (2021) considers a different model of predictions which is not directly comparable to ours. Upon the arrival of $i$-th demand point, a set of *multiple* predictions $\mathcal{S}_i$ is provided, each suggesting a list of facilities to be opened. Their algorithm achieves a cost of $O(\log(|\bigcup_i \mathcal{S}_i|) \text{OPT}(\bigcup_i \mathcal{S}_i))$ where $\text{OPT}(\bigcup_i \mathcal{S}_i)$ is the optimal using only facilities in $\bigcup_i \mathcal{S}_i$, and the algorithm also has a worst-case ratio of $O(\frac{\log n}{\log \log n})$ in case the predictions are inaccurate. Similar to the abovemention Fotakis et al. (2021a) result, this result only works for uniform opening cost while ours work for arbitrary opening costs.

## 1.2 TECHNICAL OVERVIEW

Since we aim for a worst-case (i.e., when the predictions are inaccurate) $O(\log n)$ ratio, our algorithm is based on an $O(\log n)$-competitive algorithm (that does not use predictions) by Meyerson (2001), and we design a new procedure to make use of predictions. In particular, upon the arrival of each demand point, our algorithm first runs the steps of Meyerson's algorithm , which we call the Meyerson step, and then runs a new procedure (proposed in this paper) to open additional facilities that are "near" the prediction, which we call the Prediction step.

We make use of the following crucial property of Meyerson's algorithm. Suppose before the first demand point arrives, the algorithm is already provided with an initial facility set $\bar{F}$, such that for every facility $f^*$ opened in OPT, $\bar{F}$ contains a facility $f'$ that is of distance $\eta_\infty$ to $f^*$, then Meyerson's algorithm is $O(\log \frac{n\eta_\infty}{\text{OPT}})$-competitive. To this end, our Prediction step aims to open additional facilities so that for each facility in OPT, the algorithm would soon open a close enough facility.

**Uniform opening cost.** There is a simple strategy that achieves this goal for the case of *uniform opening cost*. Whenever the Meyerson step decides to open a facility, we further open a facility at the predicted location. By doing so, we would pay twice as the Meyerson algorithm does in order to open the extra facility. As a reward, when the prediction error $\eta_\infty$ is small, the extra facility would be good and close enough to the optimal facility.

**Non-uniform opening cost.** Unfortunately, this strategy does not extend to the non-uniform case. Specifically, opening a facility exactly at the predicted location could be prohibitively expensive as it may have huge opening cost. Instead, one needs to open a facility that is "close" to the prediction, but attention must be paid to the tradeoff between the opening cost and the proximity to the prediction.

The design of the Prediction step for the non-uniform case is the most technical and challenging of our paper. We start with opening an initial facility that is far from the prediction, and then we open a series of facilities within balls (centered at the prediction) of geometrically decreasing radius, while we also allow the opening cost of the newly open facility to be doubled each time. We stop opening facilities if the total opening cost exceeds the cost incurred by the preceding Meyerson step, in order to have the opening cost bounded.

We show that our procedure always opens a "correct" facility $\hat{f}$ that is $\Theta(\eta_\infty)$ apart to the corresponding facility in OPT, and that the total cost until $\hat{f}$ is opened is merely $O(\text{OPT})$. This is guaranteed by the gradually decreasing ball radius when we build additional facilities (so that the "correct" facility will not be skipped), and that the total opening cost could be bounded by that of the last opened facility (because of the doubling opening cost). Once we open this $\hat{f}$, subsequent runs of Meyerson steps would be $O(\log \frac{n\eta_\infty}{\text{OPT}})$-competitive.

## 1.3 RELATED WORK

OFL is introduced by Meyerson (2001), where a randomized algorithm that achieves competitive ratios of $O(1)$ and $O(\log n)$ are obtained in the setting of random arrival order and adversarial arrival order, respectively. Later, Fotakis (2008) gave a lower bound of $\Omega(\frac{\log n}{\log \log n})$ and proposed a deterministic algorithm matching the lower bound. In the same paper, Fotakis also claimed that an improved analysis of Meyerson's algorithm actually yields an $O(\frac{\log n}{\log \log n})$ competitive ratio in the setting of adversarial order. Apart from this classical OFL, other variants of it are studied as well. Divéki & Imreh (2011) and Bamas et al. (2020b) considered the dynamic variant, in which an open facility can be reassigned. Another variant, OFL with evolving metrics, was studied by Eisenstat et al. (2014) and Fotakis et al. (2021b).

There is a recent trend of studying online algorithms with predictions and many classical online problems have been revisited in this framework. Lykouris & Vassilvitskii (2021) considered online caching problem with predictions and gave a formal definition of consistency and robustness. For the ski-rental problem, Purohit et al. (2018) gave an algorithm with a hyper parameter to maintain the balance of consistency and robustness. They also considered non-clairvoyant scheduling on a single machine. Gollapudi & Panigrahi (2019) designed an algorithm using multiple predictors to achieve low prediction error. Recently, Wei & Zhang (2020) showed a tight robustness-consistency trade-off for the ski-rental problem. For the caching problem, Lykouris & Vassilvitskii (2021) adapted Marker algorithm to use predictions. Following this work, Rohatgi (2020) provided an improved algorithm that performs better when the predictor is misleading. Bamas et al. (2020b) extended the online primal-dual framework to incorporate predictions and applied their framework to solve the online covering problem. Medina & Vassilvitskii (2017) considered the online revenue maximization problem and Bamas et al. (2020a) studied the online speed scaling problem. Both Antoniadis et al. (2020) and Dütting et al. (2021) considered the secretary problems with predictions. Jiang et al. (2021) studied online matching problems with predictions in the constrained adversary framework. Azar et al. (2021) and Lattanzi et al. (2020) considered flow time scheduling and online load balancing, respectively, in settings with error-prone predictions.

## 2 PRELIMINARIES

Recall that we assume a underlying metric space $(\mathcal{X}, d)$. For $S \subseteq \mathcal{X}, x \in \mathcal{X}$, define $d(x, S) := \min_{y \in S} d(x, y)$. For an integer $t$, let $[t] := \{1, 2, \ldots, t\}$. We normalize the opening cost function $w$ so that the minimum opening cost equals 1, and we round down the opening cost of each facility to the closest power of 2. This only increases the ratio by a factor of 2 which we can afford (since we make no attempt to optimize the constant hidden in big $O$). Hence, we assume the domain of $w$ is $\{2^{i-1} \mid i \in [L]\}$ for some $L$. Further, we use $G_k$ to denote the set of facilities whose opening cost is at most $2^{k-1}$, i.e. $G_k := \{f \in \mathcal{F} \mid w(f) \le 2^{k-1}\}, \forall 1 \le k \le L$. Observe that $G_0 = \emptyset$.

---

**Algorithm 1** Prediction-augmented Meyerson

---

1: initialize $F \leftarrow \emptyset$, $F_{\mathrm{P}} \leftarrow \emptyset$    $\triangleright$ $F, F_{\mathrm{P}}$ are the set of all open facilities and those opened by PRED
2: **for** arriving demand point $x \in X$ and its associated $f_x^{\mathrm{pred}}$ **do**
3:      $\mathrm{cost}_{\mathrm{M}}(x) \leftarrow \mathrm{MEY}(x)$
4:      $\mathrm{PRED}\left(f_x^{\mathrm{pred}}, \mathrm{cost}_{\mathrm{M}}(x)\right)$
5: **procedure** $\mathrm{MEY}(x)$
6:      **for** $k \in [L]$ **do**
7:          $f_k \leftarrow \arg\min_{f \in F \cup G_k} d(x, f)$
8:          $\delta_k \leftarrow d(x, f_k)$
9:          $p_k \leftarrow (\delta_{k-1} - \delta_k)/2^k$, where $\delta_0 = d(x, F)$
10:      let $s_k \leftarrow \sum_{i=k}^{L} p_i$ for $k \in [L]$
11:      sample $r \sim \mathrm{Uni}[0,1]$, and let $i \in [L]$ be the index such that $r \in [s_{i+1}, s_i)$
12:      **if** such $i$ exists **then**
13:          $F \leftarrow F \cup f_i$                                          $\triangleright$ open facility at $f_i$
14:      connect $x$ to the nearest neighbor in $F$
15:      **return** $d(x, F) + w(f_i)$              $\triangleright$ if $i$ does not exist, simply return $d(x, F)$
16: **procedure** $\mathrm{PRED}(f_x^{\mathrm{pred}}, q)$
17:      **repeat**
18:          $r \leftarrow \frac{1}{2} d(f_x^{\mathrm{pred}}, F_{\mathrm{P}})$
19:          $f_{\mathrm{open}} \leftarrow \arg\min_{f: d(f_x^{\mathrm{pred}}, f) \leq r} w(f)$ $\triangleright$ break ties by picking the closest facility to $f_x^{\mathrm{pred}}$
20:          **if** $q \geq w(f_{\mathrm{open}})$ **then**
21:              $F \leftarrow F \cup f_{\mathrm{open}}$, $F_{\mathrm{P}} \leftarrow F_{\mathrm{P}} \cup f_{\mathrm{open}}$               $\triangleright$ open facility at $f_{\mathrm{open}}$
22:              $q \leftarrow q - w(f_{\mathrm{open}})$
23:      **until** $q < w(f_{\mathrm{open}})$
24:      open facility at $f_{\mathrm{open}}$ with probability $\frac{q}{w(f_{\mathrm{open}})}$, and update $F, F_{\mathrm{P}}$

---

## 3    OUR ALGORITHM: PREDICTION-AUGMENTED MEYERSON

We prove Theorem 1.1 in this section. Our algorithm, stated in Algorithm 1, consists of two steps, the Meyerson step and the Prediction step. Upon the arrival of each demand point $x$, we first run the Meyerson algorithm Meyerson (2001), and let $\mathrm{cost}_{\mathrm{M}}(x)$ be the cost from the Meyerson step. Roughly, for a demand point $x$, the Meyerson algorithm first finds for each $k \geq 1$, a facility $f_k$ which is the nearest to $x$, among facilities of opening cost at most $2^{k-1}$ and those have been opened, and let $\delta_k = d(x, f_k)$. Then, open a random facility from $\{f_k\}_k$, such that $f_k$ is picked with probability $(\delta_{k-1} - \delta_k)/2^k$. For technical reasons, we do not normalize $\sum_k p_k$, and we simply truncate if $\sum_k p_k > 1$. Next, we pass the cost incurred in the Meyerson step (which is random) as the budget $q$ to the Prediction step, and the Prediction step would use this budget to open a series of facilities through the repeat-until loop. Specifically, for each iteration in this loop, a facility is to be opened around distance $r$ to the prediction, where $r$ is geometrically decreasing, and the exact facility is picked to be the one with the minimum opening cost. The ratio of this algorithm is stated as follows.

**Theorem 3.1.** *Prediction-augmented Meyerson is $O(\min\{\log n, \max\{1, \log \frac{n\eta_\infty}{\mathrm{OPT}}\}\})$-competitive.*

**Calibrating predictions.**    Recall that $\eta_\infty = \max_{i \in [n]} d(f_{x_i}^{\mathrm{pred}}, f_{x_i}^{\mathrm{opt}})$ can be unbounded. We show how to "calibrate" the bad predictions so that $\eta_\infty = O(\mathrm{OPT})$. Specifically, when a demand point $x$ arrives, we compute $f_x' := \arg\min_{f \in \mathcal{F}}\{d(x, f) + w(f)\}$, and we calibrate $f_x^{\mathrm{pred}}$ by letting $f_x^{\mathrm{pred}} = f_x'$ if $d(x, f_x^{\mathrm{pred}}) \geq 2d(x, f_x') + w(f_x')$.

We show the calibrated predictions satisfy $\eta_\infty = O(\mathrm{OPT})$. Indeed, for every demand point $x$, $d(f_x', f_x^{\mathrm{opt}}) \leq d(x, f_x') + d(x, f_x^{\mathrm{opt}}) \leq 2d(x, f_x') + w(f_x') \leq O(\mathrm{OPT})$, where the second to last inequality follows from the optimality (i.e., the connection cost $d(x, f_x^{\mathrm{opt}})$ has to be smaller than the cost of first opening $f_x'$ then connecting $x$ to $f_x'$). Note that, if $\eta_\infty = O(\mathrm{OPT})$ then $\frac{n\eta_\infty}{\mathrm{OPT}} = O(n)$, hence it suffices to prove a single bound $O(\max\{1, \log \frac{n\eta_\infty}{\mathrm{OPT}}\})$ for Theorem 3.1 (i.e., ignoring the outer min).

**Algorithm analysis.** Let $F_{\text{opt}}$ be the set of open facilities in the optimal solution. We examine each $f^* \in F_{\text{opt}}$ and its corresponding demand points separately, i.e. those demand points connecting to $f^*$ in the optimal solution. We denote this set of demand points by $X(f^*)$.

**Definition 3.2** (Open facilities). For every demand point $x$, let $F(x)$ be the set of open facilities right after the arrival of request $x$, $\bar{F}(x)$ be the set of open facilities right before the arrival of $x$, and $\hat{F}(x) = F(x) \setminus \bar{F}(x)$ be the set of the newly-open facilities on the arrival of $x$. Moreover, let $F_{\text{M}}(x), F_{\text{P}}(x)$ be a partition of $F(x)$, corresponding to the facilities opened by MEY and PRED, respectively. Let $\bar{F}_{\text{M}}(x), \hat{F}_{\text{M}}(x), \bar{F}_{\text{P}}(x), \hat{F}_{\text{P}}(x)$ be defined similarly.

**Definition 3.3** (Costs). Let $\text{cost}_{\text{M}}(x) = w(\hat{F}_{\text{M}}(x)) + d\left(x, \left(F_{\text{M}}(x) \cup \bar{F}_{\text{P}}(x)\right)\right)$ be the the total cost from MEY step. Let $\text{cost}_{\text{P}}(x) = w(\hat{F}_{\text{P}}(x))$ be the cost from PRED step. Let $\text{cost}(x) = \text{cost}_{\text{M}}(x) + \text{cost}_{\text{P}}(x)$ be the total cost of $x$.

Recall that after the MEY step, we assign a total budget of $\text{cost}_{\text{M}}(x)$ to the PRED step. That is, the expected cost from the PRED step is upper bounded by the cost from the MEY step. We formalize this intuition in the following lemma.

**Lemma 3.4.** *For each demand point $x \in X$, $\mathbb{E}[\text{cost}(x)] = 2\,\mathbb{E}[\text{cost}_{\text{M}}(x)]$.*

*Proof.* Consider the expected cost from the PRED step. Given arbitrary $F_p, q$, there is only one random event from the algorithm and it is straightforward to see that the expected cost equals $q$. Notice that our algorithm assigns a total budget of $q = \text{cost}_{\text{M}}(x)$. Thus, $\mathbb{E}[\text{cost}(x)] = \mathbb{E}[\text{cost}_{\text{M}}(x) + \text{cost}_{\text{P}}(x)] = 2\,\mathbb{E}[\text{cost}_{\text{M}}(x)]$. $\qquad\square$

Therefore, we are left to analyze the total expected cost from the MEY step. The following lemma is the most crucial to our analysis. Before continuing to the proof of the lemma, we explain how it concludes the proof of our main theorem.

**Lemma 3.5.** *For every facility $f^* \in F_{\text{opt}}$, we have*

$$\sum_{x \in X(f^*)} \mathbb{E}[\text{cost}_{\text{M}}(x)] \le O\left(\max\left\{1, \log\frac{n\eta_\infty}{\text{OPT}}\right\}\right) \cdot \left(w(f^*) + \sum_{x \in X(f^*)} d(x, f^*)\right) + O\left(\frac{|X(f^*)|}{n} \cdot \text{OPT}\right).$$

**Proof of Theorem 3.1.** Summing up the equation of Lemma 3.5 for every $f^* \in F_{\text{opt}}$, we have

$$\mathbb{E}[\text{ALG}] = \sum_{x \in X} \mathbb{E}[\text{cost}(x)] = 2\sum_{x \in X} \mathbb{E}[\text{cost}_{\text{M}}(x)] = 2\sum_{f^* \in F_{\text{opt}}} \sum_{x \in X(f^*)} \mathbb{E}[\text{cost}_{\text{M}}(x)]$$

$$\le \sum_{f^* \in F_{\text{opt}}} \left(O\left(\max\left\{1, \log\frac{n\eta_\infty}{\text{OPT}}\right\}\right) \cdot \left(w(f^*) + \sum_{x \in X(f^*)} d(x, f^*)\right) + O\left(\frac{|X(f^*)|}{n} \cdot \text{OPT}\right)\right)$$

$$= O\left(\max\left\{1, \log\frac{n\eta_\infty}{\text{OPT}}\right\}\right) \cdot \text{OPT},$$

where the second equality is by Lemma 3.4 and the inequality is by Lemma 3.5, and this also implies that the ratio is $O(1)$ when $\eta_\infty = 0$. $\qquad\square$

## 3.1 PROOF OF LEMMA 3.5

Fix an arbitrary $f^* \in F_{\text{opt}}$. Let $\ell^*$ be the integer such that $w(f^*) = 2^{\ell^*-1}$, or equivalently $f^* \in G_{\ell^*}$. Let $X(f^*) = \{x_1, x_2, \ldots, x_m\}$ be listed according to their arrival order. We remark that there can be other demand points arriving between $x_i$ and $x_{i+1}$, but they must be connected to other facilities in the optimal solution. Let $\ell_i$ be the index $\ell$ such that $d(f^*, \bar{F}(x_i)) \in [2^{\ell-1}, 2^\ell)$. $\ell_i$ can be negative and if $d(f^*, \bar{F}(x_i)) = 0$, let $\ell_i = -\infty$. Observe that $d(f^*, \bar{F}(x_i))$ and $\ell_i$ are non-increasing in $i$. Let $\tau$ be the largest integer $i$ with $d(f^*, \bar{F}(x_i)) > \min\{7\eta_\infty, 4w(f^*)\}$. Note that $\tau$ is a random variable. We refer to the demand sequence before $x_\tau$ as the long-distance stage and the sequence after $x_\tau$ as the short-distance stage. In this section, we focus on the case when $7\eta_\infty \le 4w(f^*)$. The proof for the other case is very similar and can be found in Section E.

**Long-distance stage.** We start with bounding the total cost in the long-distance stage. Intuitively, our procedure PRED would quickly build a facility that is close to $f^*$ within a distance of $O(\eta_\infty)$, since it is guaranteed that for each demand point $x_i$, the distance between $f^*$ and its corresponding prediction $f_{x_i}^{\text{pred}}$ is at most $\eta_\infty$. Formally, we prove the following statements.

**Lemma 3.6.** $\mathbb{E}\left[\sum_{i<\tau} \text{cost}_M(x_i)\right] \leq 2w(f^*)$.

*Proof.* The proof can be found in Section A. □

**Lemma 3.7.** $\mathbb{E}[\text{cost}_M(x_\tau)] \leq 2w(f^*) + 2\sum_{i=1}^m d(x_i, f^*)$.

*Proof.* The proof can be found in Section B. □

**Short-distance stage.** Next, we bound the costs for the subsequence of demand points whose arrival is after the event that some facility near $f^*$ is open. Formally, these demand points are $x_i$'s with $i > \tau$. The analysis of this part is conceptually similar to a part of the analysis of Meyerson's algorithm. However, ours is more refined in that it utilizes the already opened facility that is near $f^*$ to get an improved $O(\log \frac{n\eta_\infty}{\text{OPT}})$ ratio instead of $O(\log n)$. Moreover, Meyerson did not provide the full detail of this analysis, and our complete self-contained proof fills in this gap.

By the definition of $\tau$, we have $d(f^*, \bar{F}(x_i)) \leq 7\eta_\infty$ for such $i > \tau$. For integer $\ell$, let $I_\ell = \{i > \tau \mid \ell_i = \ell\}$. Then for $t$ such that $2^{t-1} > \min\{7\eta_\infty, 4w(f^*)\}$, we have $I_t = \emptyset$. Let $\underline{\ell}$ be the integer that $\frac{\text{OPT}}{n} \in [2^{\underline{\ell}-1}, 2^{\underline{\ell}})$, and let $\bar{\ell}$ be the integer such that $\min\{7\eta_\infty, 4w(f^*)\} \in [2^{\bar{\ell}-1}, 2^{\bar{\ell}})$. We partition all demand points with $i > \tau$ into groups $I_{\leq \underline{\ell}-1}, I_{\underline{\ell}}, \ldots, I_{\bar{\ell}}$ according to $\ell_i$, where $I_{\leq \underline{\ell}-1} = \bigcup_{\ell \leq \underline{\ell}-1} I_\ell$.

**Lemma 3.8.** $\mathbb{E}\left[\sum_{i \in I_{\leq \underline{\ell}-1}} \text{cost}_M(x_i)\right] \leq \frac{2|X(f^*)|}{n} \cdot \text{OPT}$.

*Proof.* The proof can be found in Section C. □

**Lemma 3.9.** *For every* $\ell \in [\underline{\ell}, \bar{\ell}]$, $\mathbb{E}\left[\sum_{i \in I_\ell} \text{cost}_M(x_i)\right] \leq 18\sum_{i \in I_\ell} d(x_i, f^*) + 32w(f^*)$.

*Proof.* The proof can be found in Section D. □

**Proof of Lemma 3.5.** We conclude Lemma 3.5 by combining Lemma 3.6, 3.7, 3.8, and 3.9.

$$
\begin{aligned}
\sum_{i \in [m]} \mathbb{E}\left[\text{cost}_M(x_i)\right] &= \mathbb{E}\left[\sum_{i \in [\tau-1]} \text{cost}_M(x_i)\right] + \mathbb{E}[\text{cost}_M(x_\tau)] \\
&\quad + \sum_{\ell=\underline{\ell}}^{\bar{\ell}} \mathbb{E}\left[\sum_{i \in I_\ell} \text{cost}_M(x_i)\right] + \sum_{i \in I_{\leq \underline{\ell}-1}} \mathbb{E}\left[\text{cost}_M(x_i)\right] \\
&\leq 2w(f^*) + \left(2w(f^*) + 2\sum_{i=1}^m d(x_i, f^*)\right) \\
&\quad + \sum_{\ell=\underline{\ell}}^{\bar{\ell}} \left(18\sum_{i \in I_\ell} d(x_i, f^*) + 32w(f^*)\right) + \frac{2|X(f^*)|}{n} \cdot \text{OPT} \\
&\leq \left(O(\bar{\ell} - \underline{\ell}) + O(1)\right) \cdot \left(\sum_{i \in [m]} d(x_i, f^*) + w(f^*)\right) + \frac{2|X(f^*)|}{n} \text{OPT} \\
&\leq O\left(\max\left\{1, \log \frac{n\eta_\infty}{\text{OPT}}\right\}\right) \cdot \left(\sum_{i \in [m]} d(x_i, f^*) + w(f^*)\right) + O\left(\frac{|X(f^*)|}{n} \text{OPT}\right).
\end{aligned}
$$

□

Table 1: Specifications of datasets

| dataset | type | size | # of dimension/edges | non-uniform |
|---------|------|------|---------------------|-------------|
| Twitter | Euclidean | 30k | # dimension = 2 | no |
| Adult | Euclidean | 32k | # dimension = 6 | no |
| US-PG | Graph | 4.9k | # edges = 6.6k | no |
| Non-Uni | Euclidean | 4.8k | # dimension = 2 | yes |

## 4 EXPERIMENTS

We validate our online algorithms on various datasets of different types. In particular, we consider three Euclidean data sets, a) Twitter (Chan et al.), b) Adult (Dua & Graff, 2017), and c) Non-Uni (Cebecauer & Buzna, 2018) which are datasets consisting of numeric features in $\mathbb{R}^d$ and the distance is measured by $\ell_2$, and one graph dataset, US-PG (Rossi & Ahmed, 2015) which represents US power grid in a graph, where the points are vertices in a graph and the distance is measured by the shortest path distance. The opening cost is non-uniform in Non-Uni dataset, while it is uniform in all the other three. These datasets have also been used in previous papers that study facility location and related clustering problems (Chierichetti et al., 2017; Chan et al., 2018; Cohen-Addad et al., 2019). A summary of the specification of datasets can be found in Table 1.

**Tradeoff between $\eta_\infty$ and empirical competitive ratio.** Our first experiment aims to measure the tradeoff between $\eta_\infty$ and the empirical competitive ratio of our algorithm, and we compare against two "extreme" baselines, a) the vanilla Meyerson's algorithm *without using* predictions which we call MEYERSON, and b) a naive algorithm that *always follows* the prediction which we call FOLLOW-PREDICT. Intuitively, a simultaneously consistent and robust online algorithm should perform similarly to the FOLLOW-PREDICT baseline when the prediction is nearly perfect, while comparable to MEYERSON when the prediction is of low quality.

Since it is NP-hard to find the optimal solution to facility location problem, we run a simple 3-approximate MP algorithm (Mettu & Plaxton, 2000) to find a near optimal solution $F^\star$ for every dataset, and we use this solution as the offline optimal solution (which we use as the benchmark for the competitive ratio). Then, for a given $\eta_\infty$ and a demand point $x$, we pick a random facility $f \in \mathcal{F}$ such that $\eta_\infty/2 \leq d(f, F_x^\star) \leq \eta_\infty$ as the prediction, where $F_x^\star$ is the point in $F^\star$ that is closest to $x$. The empirical competitive ratio is evaluated for every baselines as well as our algorithm on top of every dataset, subject to various values of $\eta_\infty$.

All our experiments are conducted on a laptop with Intel Core i7 CPU and 16GB memory. Since the algorithms are randomized, we repeat every run 10 times and take the average cost.

In Figure 1 we plot for every dataset a line-plot for all baselines and our algorithm, whose x-axis is $\eta_\infty$ and y-axis is the empirical competitive ratio. Here, the scale of x-axis is different since $\eta_\infty$ is dependent on the scale of the dataset. From Figure 1, it can be seen that our algorithm performs consistently better than MEYERSON when $\eta_\infty$ is relatively small, and it has comparable performance to MEYERSON when $\eta_\infty$ is so large that the FOLLOW-PREDICT baseline loses control of the competitive ratio. For instance, in the Twitter dataset, our algorithm performs better than MEYERSON when $\eta_\infty \leq 2$, and when $\eta_\infty$ becomes larger, our algorithm performs almost the same with MEYERSON while the ratio of FOLLOW-PREDICT baseline increases rapidly.

Furthermore, we observe that our performance lead over MERYERSON is especially significant on dataset Non-Uni whose opening cost is non-uniform. This suggests that our algorithm manages to use the predictions effectively in the non-uniform setting, even provided that the prediction is tricky to use since predictions at "good" locations could have large opening costs. In particular, our algorithm does a good job to find a "nearby" facility that has the correct opening cost and location tradeoff.

**A simple predictor and its empirical performance.** We also present a simple predictor that can be constructed easily from a training set, and evalute its performance on our datasets. We assume the predictor has access to a training set $T$. Initially, the predictor runs the 3-approximate MP algorithm on the training set $T$ to obtain a solution $F^*$. Then when demand points from the dataset (i.e.,

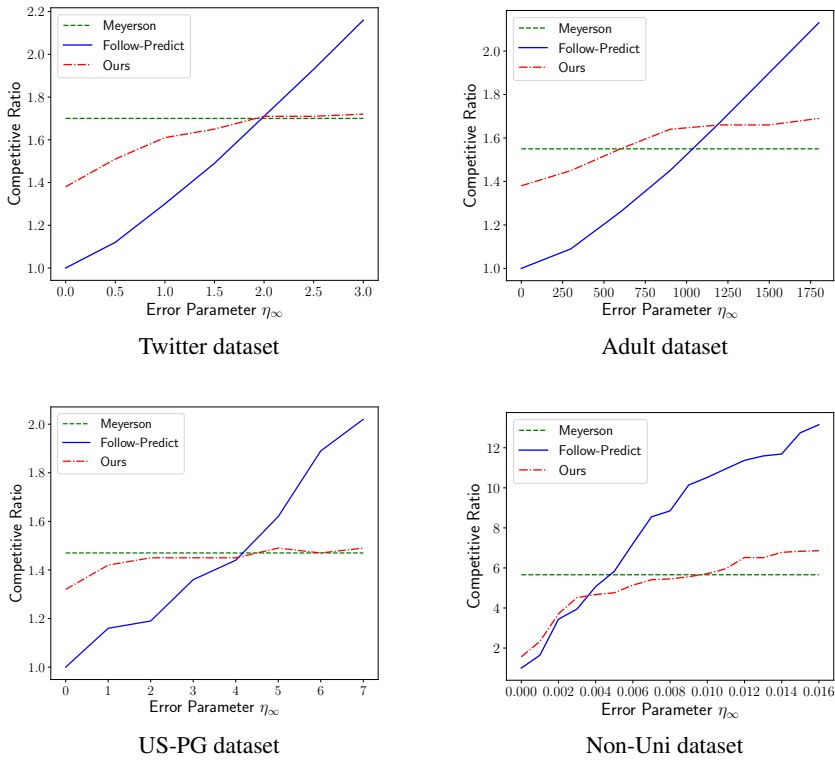

Figure 1: The tradeoff between prediction error $\eta_\infty$ and empirical competitive ratio over four datasets: Twitter, Adult, US-GD and Non-Uni.

Table 2: Empirical competitive ratio evaluation for the simple predictor

| dataset | Meyerson | Follow-Predict | Ours |
|---------|----------|----------------|------|
| Twitter | 1.70 | 1.69 | 1.57 |
| Adult | 1.55 | 1.57 | 1.49 |
| US-PG | 1.47 | 1.47 | 1.43 |
| Non-Uni | 5.66 | 5.7 | 2.93 |

test set) $X$ arrives, the predictor periodically reruns the MP algorithm, on the union of $T$ and the already-seen test data $X' \subseteq X$, and update $F^*$. For a demand point $x$, the prediction is defined as the nearest facility to $x$ in $F^*$.

To evaluate the performance of this simple predictor, we take a random sample of 30% points from the dataset as the training set $T$, and take the remaining 70% as the test set $X$. We list the accuracy achieved by the predictor in this setup in Table 2. From the table, we observe that the predictor achieves a reasonable accuracy since the ratio of Follow-Predict baseline is comparable to Meyerson's algorithm. Moveover, when combining with this predictor, our algorithm outperforms both baselines, especially on the Non-Uni dataset where the improvement is almost two times. This not only shows the effectiveness of the simple predictor, but also shows the strength of our algorithm.

Finally, we emphasize that this simple predictor, even without using any advanced machine learning techniques or domain-specific signals/features from the data points (which are however commonly used in designing predictors), already achieves a reasonable performance. Hence, we expect to see an even better result if a carefully engineered predictor is employed.

ACKNOWLEDGMENTS

This work is supported by Science and Technology Innovation 2030 — "New Generation of Artificial Intelligence" Major Project No.(2018AAA0100903), Program for Innovative Research Team of Shanghai University of Finance and Economics (IRTSHUFE) and the Fundamental Research Funds for the Central Universities. Shaofeng H.-C. Jiang is supported in part by Ministry of Science and Technology of China No. 2021YFA1000900. Zhihao Gavin Tang is partially supported by Huawei Theory Lab. We thank all anonymous reviewers for their insightful comments.

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

## A    PROOF OF LEMMA 3.6

**Lemma A.1** (Restatement of Lemma 3.6). $\mathbb{E}\left[\sum_{i<\tau} \text{cost}_{\text{M}}(x_i)\right] \leq 2w(f^*)$.

*Proof.* Let $G = \{g_1, g_2, \ldots, g_t\}$ be the facilities opened by the PRED steps on the arrivals of all $\{x_i\}_{i<\tau}$, and are listed according to their opening order. Observe that the budget that is assigned to the PRED step is $\text{cost}_{\text{M}}(x_i)$, and the expected opening cost of each PRED step equals its budget. By the definition of $G$, we have $\mathbb{E}\left[\sum_{i<\tau} \text{cost}_{\text{M}}(x_i)\right] = \mathbb{E}\left[\sum_{k\in[t]} w(g_k)\right]$.

Next, we prove that the opening costs of $g_k$'s are always strictly increasing. For each $k \in [t-1]$, suppose $g_k$ is opened on the arrival of demand $x$ and $g_{k+1}$ is opened on the arrival of demand $y$. Let $F$ denote the set of all open facilities right before $g_{k+1}$ is opened. Let

$f := \arg\min_{f:d(f_y^{\mathrm{pred}},f)\le\frac{1}{2}d(f_y^{\mathrm{pred}},g_k)} w(f)$, we must have $w(g_{k+1}) \ge w(f)$, since $g_{k+1} = \arg\min_{f:d(f_y^{\mathrm{pred}},f)\le\frac{1}{2}d(f_y^{\mathrm{pred}},F)} w(f)$ and that $d(f_y^{\mathrm{pred}}, F) \le d(f_y^{\mathrm{pred}}, g_k)$. Moreover,

$$d(f, f_x^{\mathrm{pred}}) \le d(f, f_y^{\mathrm{pred}}) + d(f_x^{\mathrm{pred}}, f_y^{\mathrm{pred}}) \le \frac{1}{2}d(f_y^{\mathrm{pred}}, g_k) + 2\eta_\infty$$

$$\le \frac{1}{2}\left(d(f_x^{\mathrm{pred}}, g_k) + d(f_x^{\mathrm{pred}}, f_y^{\mathrm{pred}})\right) + 2\eta_\infty \le \frac{1}{2}d(f_x^{\mathrm{pred}}, g_k) + 3\eta_\infty < d(f_x^{\mathrm{pred}}, g_k).$$

Here, the second and fourth inequalities use the fact that $d(f_x^{\mathrm{pred}}, f_y^{\mathrm{pred}}) \le d(f_x^{\mathrm{pred}}, f^*) + d(f^*, f_y^{\mathrm{pred}}) \le 2\eta_\infty$ since both $x, y \in X(f^*)$. The last inequality follows from the fact that $d(f_x^{\mathrm{pred}}, g_k) \ge d(g_k, f^*) - d(f_x^{\mathrm{pred}}, f^*) > 7\eta_\infty - \eta_\infty = 6\eta_\infty$ by the definition of $\tau$.

Consider the moment when we open $g_k$, the fact that we do not open facility $f$ implies that $w(g_k) < w(f)$. Therefore, $w(g_k) < w(f) \le w(g_{k+1})$. Recall that the opening cost of each facility is a power of 2. We have that $\sum_{k\in[t]} w(g_k) \le 2w(g_t)$.

Finally, we prove that the opening cost $w(g_t)$ is at most $w(f^*)$. Consider the moment when $g_t$ is open, let $x$ be the corresponding demand point. Notice that

$$d(f^*, f_x^{\mathrm{pred}}) \le \eta_\infty < \frac{1}{2}d(f^*, \bar{F}(x)) \le \frac{1}{2}d(f^*, \bar{F}_\mathrm{P}(x)),$$

by the definition of $\tau$. However, we open $g_t$ instead of $f^*$ on Line 20 of our algorithm. It must be the case that $w(f^*) \ge w(g_t)$.

To conclude the proof, we have $\sum_{k\in[t]} w(g_k) \le 2w(g_t) \le 2w(f^*)$. $\qquad\square$

## B  PROOF OF LEMMA 3.7

**Lemma B.1** (Restatement of Lemma 3.7). $\mathbb{E}[\mathrm{cost}_\mathrm{M}(x_\tau)] \le 2w(f^*) + 2\sum_{i=1}^{m} d(x_i, f^*)$.

*Proof.* We bound the opening cost and connection cost separately.

**Opening cost.**  By the algorithm, we know for every $1 \le i \le m$,

$$\mathbb{E}\left[w(\hat{F}_\mathrm{M}(x_i))\mathbf{1}(w(\hat{F}_\mathrm{M}(x_i)) > w(f^*))\right] \le \sum_{i>l^*} \frac{\delta_{i-1} - \delta_i}{2^i} \cdot 2^i = \delta_{l^*} \le d(x_i, f^*).$$

Hence, using the fact that at most one facility is opened by Meyerson step for each demand,

$$\mathbb{E}[w(\hat{F}_\mathrm{M}(x_\tau))] \le \mathbb{E}\left[\max_{f\in F_\mathrm{M}(x_m)} w(f)\right] \le w(f^*) + \mathbb{E}\left[\sum_{f\in F_\mathrm{M}(x_m), w(f)>w(f^*)} w(f)\right]$$

$$\le w(f^*) + \sum_{i=1}^{m} d(x_i, f^*).$$

This finishes the analysis for the opening cost.

**Connection cost.**  We claim that $d(x_\tau, F(x_\tau)) \le w(f^*) + d(x_\tau, f^*)$ (with probability 1).

Let $\delta_0, \delta_1, \ldots, \delta_L$ be defined as in our algorithm. We assume $\delta_0 = d(x_\tau, \bar{F}(x_\tau)) > w(f^*) + d(x_\tau, f^*)$, as otherwise the claim holds immediately. Let $k := \max\{k : \delta_k > w(f^*) + d(x_\tau, f^*)\}$. Then

$$\Pr[d(x_\tau, F(x_\tau)) \le w(f^*) + d(x_\tau, f^*)]$$

$$= \min\left(\sum_{i=k+1}^{L} \frac{\delta_{i-1} - \delta_i}{2^i}, 1\right) \ge \min\left(\sum_{i=k+1}^{\ell^*} \frac{\delta_{i-1} - \delta_i}{2^{\ell^*}}, 1\right) = \min\left(\frac{\delta_k - \delta_{\ell^*}}{2^{\ell^*}}, 1\right) = 1.$$

Therefore, $\mathbb{E}[\mathrm{cost}_\mathrm{M}(x_\tau)] = \mathbb{E}[w(\hat{F}_\mathrm{M}(x_\tau))] + \mathbb{E}[d(f^*, F(x_\tau))] \le 2w(f^*) + 2\sum_{i=1}^{m} d(x_i, f^*)$, which concludes the proof. $\qquad\square$

## C PROOF OF LEMMA 3.8

**Lemma C.1** (Restatement of Lemma 3.8). $\mathbb{E}\left[\sum_{i \in I_{\leq \ell-1}} \mathrm{cost}_{\mathrm{M}}(x_i)\right] \leq \frac{2|X(f^*)|}{n} \cdot \mathrm{OPT}$.

*Proof.* For $i \in I_{\leq \ell-1}$, we have that its connection cost $d(x_i, F(x_i)) \leq d(x_i, \bar{F}(x_i)) \leq 2^{\ell-1} \leq \frac{\mathrm{OPT}}{n}$. Let $\delta_0, \delta_1, \ldots, \delta_L$ and $f_1, \ldots, f_L$ be defined as in our algorithm upon $x_i$'s arrival. Then, its expected opening cost in the MEY step equals

$$\mathbb{E}\left[w(\hat{F}_{\mathrm{M}}(x_i))\right] = \sum_{k \in [L]} \Pr\left[\hat{F}_{\mathrm{M}}(x_i) = \{f_k\}\right] \cdot w(f_k) \leq \sum_{k \in [L]} \min\left(p_k, 1\right) \cdot 2^{k-1}$$

$$\leq \sum_{k \in [L]} \min\left(\frac{\delta_{k-1} - \delta_k}{2^k}, 1\right) \cdot 2^{k-1} \leq \sum_{k \in [L]} \frac{\delta_{k-1} - \delta_k}{2} \leq \frac{\delta_0}{2} \leq \frac{\mathrm{OPT}}{2n}.$$

To sum up,

$$\mathbb{E}\left[\sum_{i \in I_{\leq \ell-1}} \mathrm{cost}_{\mathrm{M}}(x_i)\right] \leq \mathbb{E}\left[\sum_{i \in I_{\leq \ell-1}} \left(d(x_i, F(x_i)) + w(\hat{F}_{\mathrm{M}}(x_i))\right)\right] \leq \sum_{i \in I_{\leq \ell-1}} \frac{3}{2n} \mathrm{OPT}$$

$$< \frac{2|X(f^*)|}{n} \cdot \mathrm{OPT}.$$

$\square$

## D PROOF OF LEMMA 3.9

**Lemma D.1** (Restatement of 3.9). *For every $\ell \in [\underline{\ell}, \bar{\ell}]$,*

$$\mathbb{E}\left[\sum_{i \in I_\ell} \mathrm{cost}_{\mathrm{M}}(x_i)\right] \leq 18 \sum_{i \in I_\ell} d(x_i, f^*) + 32w(f^*).$$

*Proof.* Recall that we relabeled and restricted the indices of points in $X$ to $X(f^*) = \{x_1, \ldots, x_m\}$. However, in this proof, we also need to talk about the other points in the original data set (that do not belong to $X(f^*)$). Hence, to avoid confusions, we rewrite $X = \{y_1, \ldots, y_n\}$ (so $y_1, \ldots, y_n$ are the demand points in order), and we define $\sigma : [m] \to [n]$ that maps elements $x_j \in X(f^*)$ to its identity $y_i \in X$, i.e., $\sigma(j) = i$.

For $i \in [n]$, let

$$z_i^{\mathrm{M}} := \begin{cases} \hat{F}_{\mathrm{M}}(y_i) & \hat{F}_{\mathrm{M}}(y_i) \neq \emptyset \\ \mathrm{none} & \mathrm{otherwise} \end{cases},$$

and define $z_i^{\mathrm{P}}$ similarly. Since the MEY step opens at most one facility, $z_i^{\mathrm{M}}$ is either a singleton or "none". Finally, define $z_i := z_i^{\mathrm{M}} \cup z_i^{\mathrm{P}}$.

Define $\tau_\ell = \min\{i \in [n] \mid z_i^{\mathrm{M}} \neq \mathrm{none} \text{ and } d(f^*, z_i^{\mathrm{M}}) < 2^{\ell-1}\}$, we have

$$\mathbb{E}\left[\sum_{i \in I_\ell} \mathrm{cost}_{\mathrm{M}}(x_i) - 18d(x_i, f^*)\right] = \mathbb{E}\left[\sum_{i \in [\tau_\ell]} (\mathrm{cost}_{\mathrm{M}}(y_i) - 18d(y_i, f^*)) \mathbf{1}(i \in \sigma(I_\ell))\right]. \quad (1)$$

Next, we prove a stronger version by induction, and the induction hypothesis goes as follows. For every $j \in [n]$,

$$\mathbb{E}\left[\sum_{i=j}^{\tau_\ell} (\mathrm{cost}_{\mathrm{M}}(y_i) - 18d(y_i, f^*)) \mathbf{1}(i \in \sigma(I_\ell)) \mid \mathcal{E}_{j-1}\right] \leq 32w(f^*),$$

where $\mathcal{E}_j = (z_1, \ldots, z_j)$. Clearly, applying this with $j = 1$ implies (1), hence it suffices to prove this hypothesis.

**Base case.** The base case is $j = n$, and we would do a backward induction on $j$. Since we condition on $\mathcal{E}_{j-1}$, the variables $\delta_0, \ldots, \delta_L$ in the MEY step of $y_j$, as well as wether or not $j \in \sigma(I_\ell)$, are fixed. Hence,

$$\mathbb{E}[\text{cost}_\text{M}(y_j) \mid \mathcal{E}_{j-1}] \le d(y_j, \bar{F}(y_j)) + \sum_{i=1}^{L} \frac{\delta_{i-1} - \delta_i}{2^i} \cdot 2^i \le 2d(y_j, \bar{F}(y_j)).$$

This implies

$$\mathbb{E}\left[\sum_{i=j}^{\tau_\ell} (\text{cost}_\text{M}(y_i) - 18d(y_i, f^*)) \mathbf{1}(i \in \sigma(I_\ell)) \mid \mathcal{E}_{j-1}\right]$$

$$\le \mathbb{E}[(\text{cost}_\text{M}(y_n) - 18d(y_n, f^*))\mathbf{1}(n \in \sigma(I_\ell)) \mid \mathcal{E}_{n-1}]$$

$$\le 2d(y_n, \bar{F}(y_n)) - 18d(y_n, f^*) \mid_{n \in \sigma(I_\ell)}$$

$$\le 2d(y_n, \bar{F}(y_n)) - 2d(y_n, f^*) \mid_{n \in \sigma(I_\ell)}$$

$$\le 2d(\bar{F}(y_n), f^*) \mid_{n \in \sigma(I_\ell)}$$

$$\le 2^{\ell+1} \le 2^{\ell^*+1} \le O(w(f^*)).$$

**Inductive step.** Next, assume the hypothesis holds for $j + 1, j + 2, \ldots, n$, and we would prove the hypothesis for $j$. We proceed with the following case analysis.

**Case 1:** $j \notin \sigma(I_\ell)$. We do a conditional expectation argument, and we have

$$\mathbb{E}\left[\sum_{i=j}^{\tau_\ell} (\text{cost}_\text{M}(y_i) - 18d(y_i, f^*)) \mathbf{1}(i \in \sigma(I_\ell)) \mid \mathcal{E}_{j-1}\right]$$

$$= \sum_{z_j^\text{M}, z_j^\text{P}} \Pr\left[z_j^\text{M}, z_j^\text{P} \mid \mathcal{E}_{j-1}\right] \mathbb{E}\left[\sum_{i=j}^{\tau_\ell} (\text{cost}_\text{M}(y_i) - 18d(y_i, f^*)) \mathbf{1}(i \in \sigma(I_\ell)) \mid \mathcal{E}_{j-1}, z_j^\text{M}, z_j^\text{P}\right]$$

$$\le 32w(f^*).$$

where the second step follows by induction hypothesis.

**Case 2:** $j \in I_\ell$ and $8d(y_j, f^*) > d(f^*, \bar{F}(y_j))$. We have
$$d(y_j, F(y_j)) \le d(y_j, f^*) + d(f^*, F(y_j)) \le 9d(y_j, f^*).$$

Hence $\mathbb{E}[\text{cost}_\text{M}(y_j) - 18d(y_j, f^*)] \le 0$, and the hypothesis follows from a similar argument as in Case 1.

**Case 3:** $j \in I_\ell$, and $8d(y_j, f^*) \le d(f^*, \bar{F}(y_j))$. We have

$$\delta_{\ell^*} \le d(y_j, f^*) \le \frac{1}{8}d(f^*, \bar{F}(y_j)) \le 2^{\ell-3}.$$

Let $D = \{f : d(f^*, f) \ge 2^{\ell-1}\} \cup \{\text{none}\}$. Let $k = \min\{k' \mid \delta_{k'} \le 2^{\ell-2}\}$. Then $\forall i \ge k$, $d(f_i, f^*) \le d(y_j, f_i) + d(y_j, f^*) < 2^{\ell-1}$.

$$\Pr\left[\tau_\ell \le j \mid \mathcal{E}_{j-1}\right] \ge \Pr\left[z_j^\text{M} \notin D \mid \mathcal{E}_{j-1}\right] \ge \Pr\left[f_i \text{ is open for some } i \ge k \mid \mathcal{E}_{j-1}\right]$$

$$= \min\left(\sum_{i \ge k} p_i, 1\right) = \min\left(\sum_{i \ge k} \frac{\delta_{i-1} - \delta_i}{2^i}, 1\right)$$

$$\ge \min\left(\frac{\delta_{k-1} - \delta_{\ell^*}}{2^{\ell^*}}, 1\right) \ge \min\left(\frac{2^{\ell-2} - 2^{\ell-3}}{2^{\ell^*}}, 1\right) \ge \min\left(2^{\ell-3-\ell^*}, 1\right)$$

$$\ge \frac{\delta_0}{16w(f^*)} \ge \frac{\mathbb{E}[\text{cost}_\text{M}(y_j) \mid \mathcal{E}_{j-1}]}{32w(f^*)},$$

where the second last inequality follows from $\delta_0 = d(y_j, \bar{F}(y_j)) \le 2^\ell$ and $w(f^*) = 2^{\ell^*-1}$. Thus,

$$\text{LHS} = \mathbb{E}[\text{cost}_{\mathrm{M}}(y_j) \mid \mathcal{E}_{j-1}]$$

$$+ \sum_{t_j \in D} \sum_{z_j^{\mathrm{P}}} \Pr[z_j^{\mathrm{M}} = t_j, z_j^{\mathrm{P}} \mid \mathcal{E}_{j-1}] \, \mathbb{E}\left[ \sum_{i=j+1}^{\tau_\ell} (\text{cost}_{\mathrm{M}}(y_i) - 18d(y_i, f^*)) \mathbf{1}(i \in \sigma(I_\ell)) \mid \mathcal{E}_{j-1}, z_j^{\mathrm{M}} = t_j, z_j^{\mathrm{P}} \right]$$

$$\le \Pr\left[ z_j^{\mathrm{M}} \notin D \mid \mathcal{E}_{j-1} \right] \cdot 32w(f^*) + \Pr\left[ z_j^{\mathrm{M}} \in D \mid \mathcal{E}_{j-1} \right] \cdot 32w(f^*)$$

$$\le 32w(f^*),$$

where the second step follows by induction hypothesis. In summary, we have

$$\mathbb{E}\left[ \sum_{i=1}^{\tau_\ell} (\text{cost}_{\mathrm{M}}(y_i) - 18d(y_i, f^*)) \mathbf{1}(i \in \sigma(I_\ell)) \mid \mathcal{E}_{j-1} \right] \le 32w(f^*)$$

$\square$

# E   Proof of Lemma 3.5: When $7\eta_\infty > 4w(f^*)$

The proof is mostly the same as in the other case which we prove in Section 3.1, and we only highlight the key differences. We use the same definition for parameters $\tau$, $\underline{\ell}$, and $\bar{\ell}$. It can be verified that Lemma 3.7, 3.8 and 3.9 still holds and their proof in Section 3.1 still works.

However, Lemma 3.6 relies on that $7\eta_\infty \le 4w(f^*)$, which does not work in the current case. We provide Lemma E.2 in replacement of Lemma 3.6, which offers a slightly different bound but it still suffices for Lemma 3.5.

**Lemma E.1.** *For every $i < \tau$, $d(x_i, f^*) \ge w(f^*)$.*

*Proof.* We prove the statement by contradiction. Suppose $d(x_i, f^*) < w(f^*)$. Let $\delta_0, \delta_1, \ldots, \delta_W$ be defined as in our algorithm. Then, we have

$$\delta_0 = d(x_i, \bar{F}(x_i)) \ge d(f^*, \bar{F}(x_i)) - d(f^*, x_i) \ge d(f^*, \bar{F}(x_i)) - w(f^*) > 3w(f^*),$$

where the last inequality follows from the definition of $\tau$ that $d(f^*, \bar{F}(x_i)) > 4w(f^*)$. Let $k = \min\{k \mid \delta_k \le 3w(f^*)\}$. We have $k \ge 1$. We prove that a facility within a distance of $3w(f^*)$ from $x_i$ must be open at this step.

$$\Pr\left[ d(x_i, F(x_i)) \le 3w(f^*) \right] \ge \Pr\left[ f_j \text{ is opened for some } j \ge k \right]$$

$$\ge \min\left( \sum_{j \ge k} p_k, 1 \right) \ge \min\left( \sum_{j=k}^{\ell^*} \frac{\delta_{j-1} - \delta_j}{2^j}, 1 \right)$$

$$\ge \min\left( \frac{\delta_{k-1} - \delta_{\ell^*}}{2^{\ell^*}}, 1 \right) \ge \min\left( \frac{3w(f^*) - w(f^*)}{2^{\ell^*}}, 1 \right) = 1,$$

where the last inequality uses the fact that $\delta_{\ell^*} \le d(x_i, f^*) \le w(f^*)$. Consequently,

$$d(f^*, \hat{F}(x_{i+1})) \le d(f^*, F(x_i)) \le d(x_i, F(x_i)) + d(x_i, f^*) < 4w(f^*),$$

which contradicts the definition of $\tau$. $\square$

**Lemma E.2.** $\mathbb{E}\left[ \sum_{i \in [\tau]} \text{cost}(x_i) \right] \le 6 \, \mathbb{E}[\sum_{i \in [\tau]} d(x_i, f^*)] + 4 \cdot w(f^*)$.

*Proof.* On the arrival of $x_i$ for each $i \in [\tau]$, let $\delta_0, \delta_1, \ldots, \delta_L$ be defined as in our algorithm. We first study the connecting cost of request $x_i$. We have

$$d(x_i, F(x_i)) \le \max\left( d(x_i, F(x_j)\backslash\hat{F}(x_i)), d(x_i, \hat{F}(x_i)) \right) = \max\left( \delta_0, d(x_i, \hat{F}(x_i)) \right).$$

We prove this value is at most $d(x_i, f^*) + 2w(f^*)$. Suppose $\delta_0 > d(x_i, f^*) + 2w(f^*)$, as otherwise the statement holds. Recall that $\delta_j$ is non-increasing, let $k = \min\{k \mid \delta_k \leq d(x_i, f^*) + 2w(f^*)\}$. By the definition of $k$, we have that

$$\Pr\left[d(x_i, F(x_i)) \leq d(x_i, f^*) + 2w(f^*)\right] \geq \Pr[f_j \text{ is opened for some } j \geq k]$$

$$\geq \min\left(\sum_{j \geq k} p_j, 1\right) \geq \min\left(\sum_{j \geq k}^{\ell^*} \frac{\delta_{j-1} - \delta_j}{2^j}, 1\right) \geq \min\left(\frac{\delta_{k-1} - \delta_{\ell^*}}{2^{\ell^*}}, 1\right)$$

$$\geq \min\left(\frac{d(x_i, f^*) + 2w(f^*) - \delta_{\ell^*}}{2^{\ell^*}}, 1\right) \geq \min\left(\frac{w(f^*)}{2^{\ell^* - 1}}, 1\right) = 1.$$

To sum up, we have shown that the connecting cost $d(x_i, F(x_i)) \leq d(x_i, f^*) + 2w(f^*)$.

Next, we study the expected opening cost $\mathbb{E}[w(\hat{F}(x_i))]$. We have

$$\mathbb{E}[w(\hat{F}(x_i))] = \sum_{j \in [L]} \Pr\left[\hat{F}(x_i) = \{f_j\}\right] \cdot w(f_j) \leq \sum_{j \in [L]} \min\left(\frac{\delta_{j-1} - \delta_j}{2^j}, 1\right) \cdot 2^{j-1}$$

$$\leq \sum_{j \leq \ell^*} 2^{j-1} + \sum_{j > \ell^*} \frac{\delta_{j-1} - \delta_j}{2^j} \cdot 2^{j-1} < 2^{\ell^*} + \frac{1}{2}\delta_{\ell^*} < 2w(f^*) + d(x_i, f^*).$$

By Lemma E.1, we conclude the proof of the statement:

$$\mathbb{E}\left[\sum_{i \in [\tau]} \text{cost}(x_i)\right] = \mathbb{E}\left[\sum_{i \in [\tau]} \left(d(x_i, F(x_i)) + w(\hat{F}(x_i))\right)\right]$$

$$\leq \mathbb{E}\left[\sum_{i \in [\tau]} (2d(x_i, f^*) + 4w(f^*))\right]$$

$$\leq 6\,\mathbb{E}\left[\sum_{i < \tau} d(x_i, f^*) + (2d(x_\tau, f^*) + 4w(f^*))\right]$$

$$\leq 6\,\mathbb{E}\left[\sum_{i \in [\tau]} d(x_i, f^*)\right] + 4w(f^*).$$

$\square$

# F  LOWER BOUND

In the classical online facility location problem, a tight lower bound of $O(\frac{\log n}{\log \log n})$ is established by Fotakis (2008), even for the special case when the facility cost is uniform and the metric space is a binary hierarchically well-separated tree (HST). We extend their construction to the setting with predictions, proving that when the predictions are not precise (i.e. $\eta_\infty > 0$), achieving a competitive ratio of $o(\frac{\log \frac{n\eta_\infty}{\text{OPT}}}{\log \log n})$ is impossible.

**Theorem F.1.** *Consider OFL with predictions with a uniform opening cost of 1. For every $\eta_\infty \in (0, 1]$, there exists a class of inputs, such that no (randomized) online algorithm is $o(\frac{\log \frac{n\eta_\infty}{\text{OPT}}}{\log \log n})$-competitive, even when $\eta_1 = O(1)$.*

Before we provide the proof of our theorem, we give some implications of our theorem. First of all, we normalize the opening cost to be 1. Consequently, the optimal cost is at least 1. Furthermore, we are only interested in the case when $\eta_\infty \leq 1$. Indeed, the optimal facility for each demand point must be within a distance of 1, as otherwise, we can reduce the cost by opening a new facility at the demand point. Therefore, we can without loss of generality to study predictions with $\eta_\infty = O(1)$.

Without the normalization, our theorem implies an impossibility result of $o(\frac{\log \frac{n\eta_\infty}{\text{OPT}}}{\log \log n})$ for online facility location with predictions.

Moreover, recall the definition of total prediction error $\eta_1 = \sum_{i=1}^{n} d(f_{x_i}^{\mathrm{pred}}, f_{x_i}^{\mathrm{opt}})$, which is at least $\eta_\infty = \max_i d(f_{x_i}^{\mathrm{pred}}, f_{x_i}^{\mathrm{opt}})$. Our theorem states that even when the total error $\eta_1$ is constant times larger than the opening cost of 1, there is no hope for a good algorithm. Note that our bound above holds for any constant value of $\eta_\infty$. As an implication, our construction rules out the possibility of an $o(\frac{\log \frac{\eta_1}{\mathrm{OPT}}}{\log \log n})$-competitive algorithm.

*Proof.* By Yao's principle, the expected cost of a randomized algorithm on the worst-case input is no better than the expected cost for a worst-case probability distribution on the inputs of the deterministic algorithm that performs best against that distribution. For each $\eta_\infty$, we shall construct a family of randomized instance, so that the expected cost of any deterministic algorithm is at least $\Omega(\frac{\log \frac{n\eta_\infty}{\mathrm{OPT}}}{\log \log n})$-competitive.

We first import the construction for the classical online facility location problem by Fotakis (2008). Consider a hierarchically well-separated perfect binary tree. Let the distance between root and its children as $D$. For every vertex $i$ of the tree, the distance between $i$ and its children is $\frac{1}{m}$ times the distance between $i$ and its parent. That is to say, the distance between a height $i$ vertex and its children of height $i+1$ is $\frac{D}{m^i}$. Let the height of the tree be $h$.

The demand sequence is consisted of $h+1$ phases. For each $0 \le i \le h$, in the $(i+1)$-th phase, $m^i$ demand points arrive consecutively at some vertex of height $i$. For $i \ge 1$, the identity of the height $i$ vertex is independently and uniformly chosen between the two children of $i$-th phase vertex.

Consider the solution that opens only one facility at the leaf node that is the $(h+1)$-th phase demand vertex and assign all demand points to the leaf. The cost of this solution equals

$$1 + \sum_{i=0}^{h} m^i \sum_{j=i}^{h-1} \frac{D}{m^j} \le 1 + \sum_{i=0}^{h-1} m^i \frac{D}{m^i} \frac{m}{m-1} = 1 + hD\frac{m}{m-1}.$$

This value serves as an upper bound of the optimal cost. I.e., $\mathrm{OPT} \le 1 + hD\frac{m}{m-1}$.

Next, we describe the predictions associated with the demand points. Intuitively, we try the best to hide the identity of the leaf node in the last phase. We denote the leaf node as $f^*$, which is also the open facility in the above described solution. Our prediction is produced according to the following rule. When the distance between the current demand point and $f^*$ is less than $\eta_\infty$, let the prediction be the same as the demand vertex. Otherwise, let the prediction be the vertex on the path from the current demand point to $f^*$, whose distance to $f^*$ equals $\eta_\infty$.

We prove a lower bound on the expected cost of any deterministic algorithm. We first overlook the first a few phases until the subtree of the current demand vertex has diameter less than $\eta_\infty$. Let it be the $h'$-th phase. We now focus on the cost induced after the $h'$-th phase and notice that the predictions are useless as they are just the demand points. When the $m^i$ demand points at vertex of height $i$ comes, we consider the following two cases:

**Case 1:** There is no facility opened in the current subtree. Then we either open a new facility bearing an opening cost 1 or assign the current demands to facilities outside the subtree bearing a large connection cost at least $\frac{D}{m^{i-1}}$ per demand. So the cost of this vertex is $\min\{1, Dm\}$.

**Case 2:** There has been at least one facility opened in the current subtree. Since the next vertex is uniformly randomly chosen in its two children, the expected number of facility that will not enter the next phase subtree is at least $\frac{1}{2}$. We call it abandoned. The expected sum of abandoned facility is $\frac{1}{2}$ of the occurrence of case 2. Thus the expected cost in every occurrence of case 2 is $\frac{1}{2}$.

We carefully choose $D, m, h$ so that 1) $mD = 1$; 2) the total number of demand points is $n$, i.e. $\sum_{i=0}^{h} m^i = n$; and 3) $\frac{D}{m^{h'}} = \eta_\infty$. These conditions give that $(h+1)\log m = \log n$, $(h'+1)\log m = -\log \eta_\infty$. Consequently, $h - h' = \frac{\log n\eta_\infty}{\log m}$.

To sum up, a lower bound of any deterministic algorithm is $(h-h')\min\{\frac{1}{2}, mD\} = \Omega\left(\frac{\log n\eta_\infty}{\log m}\right)$, while the optimal cost is at most $1 + hD\frac{m}{m-1} = O\left(\frac{\log n}{m \log m}\right)$. Setting $m = \frac{\log n}{\log \log n}$, the optimal

cost is $O(1)$. And we prove the claimed lower bound of $\Omega(\frac{\log n\eta_\infty}{\log\log n})$. Finally, it is straightforward to see that the summation of the prediction error $\eta_1 \leq \text{OPT} = O(1)$. $\qquad\square$

## G $t$-OUTLIER SETTING

Observe that our main error parameter $\eta_\infty$ could be very sensitive to a single outlier prediction that has a large error. To make the error parameter bahave more smoothly, we introduce an integer parameter $1 \leq t \leq n$, and define $\eta_\infty^{(t)}$ as the $t$-th largest prediction error, i.e., the maximum prediction error excluding the $t - 1$ high-error outliers. Define $\eta_1^{(t)}$ similarly.

In the following, stated in Theorem G.1, we argue that our algorithm, without any modification (but with an improved analysis), actually has a ratio of $O(\log(1 + t) + \log \frac{n\eta_\infty^{(t)}}{\text{OPT}})$ that holds for *every* $t$. This also means the ratio would be $\min_{1 \leq t \leq n} O(\log(1 + t) + \log \frac{n\eta_\infty^{(t)}}{\text{OPT}})$, and this is clearly a generalization of Theorem 1.1, since $\eta_\infty^{(t)} = \eta_\infty$ when $t = 1$.

Moreover, we note that our lower bound, Theorem 1.2, is still valid for ruling out the possibility of replacing $\eta_\infty^{(t)}$ with $\eta_1^{(t)}$ in the abovementioned ratio, since one can still apply Theorem 1.2 with $t = 1$. However, it is an interesting open question to explore whether or not an algorithm with a ratio like $O(\log t + \frac{\eta_1^{(t)}}{\text{OPT}})$ for some *specific* $t$ (e.g., $t \geq \sqrt{n}$) exists.

**Theorem G.1.** *For every integer* $1 \leq t \leq n$, *Algorithm* 1 *is* $O(\log(t + 1) + \min\{\log n, \max\{1, \log \frac{n\eta_\infty^{(t)}}{\text{OPT}}\}\})$-*competitive.*

*Proof sketch.* Let $\Gamma = (\gamma_1, \gamma_2, \ldots, \gamma_{t-1})$ be indices of the $t - 1$ largest prediction error $d(f_{x_\gamma}^{\text{pred}}, f_{x_\gamma}^{\text{opt}})$. The high level idea is to break the dataset $X$ into a good part $X_G := \{x_i : i \in [n]\backslash\Gamma\}$ and a bad part $X_B := \{x_i : i \in \Gamma\}$ according to whether or not the prediction is within the outlier, and we argue that the expected cost of ALG on $X_G$ is $O(\frac{n\eta_\infty^{(t)}}{\text{OPT}})$ times larger than that in OPT, and show the expected cost on $X_B$ is $O(\log(t + 1))$ times.

For the good part $X_G$, we let $X_-(f^*) := X(f^*)\backslash\Gamma$ and apply a similar analysis as in the proof of Lemma 3.5 to show that

$$E\left[\sum_{x \in X_-(f^*)} \text{cost}(x)\right] \leq O\left(\max\left\{1, \log \frac{n\eta_\infty^{(t)}}{\text{OPT}}\right\}\right)\left(w(f^*) + \sum_{x \in X_-(f^*)} d(x, f^*)\right).$$

For the bad part $X_B$, we assume all predictions for points in $X_B$ are of infinite error (which could only increase the cost of the algorithm), and let $X_+(f^*) := X(f^*) \cap \Gamma$. Then this lies in the case of Section E, where $7\eta_\infty > 4w(f^*)$, which essentially means the prediction is almost useless and the whole proof reverts to pure Meyerson's algorithm. Combine the arguments from Section E and the analysis for the short distance stage, we have

$$E\left[\sum_{x \in X_+(f^*)} \text{cost}(x)\right] \leq O(\log(t + 1))\left(w(f^*) + \sum_{x \in X_+(f^*)} d(x, f^*)\right).$$

We finish the proof by combining the bound for the two parts. $\qquad\square$

