# OpenReview forum: "Online Facility Location with Predictions"
_ICLR.cc/2022/Conference — ICLR 2022 Poster_

### Official Review · Reviewer_m2EN · 2021-10-30

**Correctness:** 3
**Technical Novelty And Significance:** 3
**Empirical Novelty And Significance:** 2
**Recommendation:** 6
**Confidence:** 4

**Main Review:**

The main contribution of the paper is giving a near-optimal result for online facility location with predictions, along with a nearly-matching lower bound. The only other result is the Fotakis et al. paper the authors discuss in the intro, which only considers the setting where all facilities have opening cost 1, whereas this paper handles non-uniform opening costs; even ignoring the correctness issues of the Fotakis et al. paper brought up by the authors, the algorithm's prediction step is to the best of my knowledge a novel one, i.e. it is not simply lifting a technique from another paper, and the difference between the weighted and unweighted setting is a key technical challenge in their results. So I would view this as a substantial improvement over the Fotakis et al. paper, even if the Fotakis et al. paper is correct/gets a better competitive ratio in the setting where $\eta_1$ is small (although I agree with the authors that the bound claimed in the Fotakis et al. paper is somewhat fishy due to the fact that it could be negative when $\eta_1$ is large compared to $\eta_\infty$, and was not able to discern from reading that paper where the catch is). Furthermore, the design of the prediction step is fairly non-trivial to me, but after reading the paper I felt I had some good intuition for why this was the right way to use the predicted facilities, i.e. that the prediction step had a nice intuitive explanation. The experiments also nicely supplement the theoretical results. While the lower bound construction is not particularly novel by itself (I believe it is a modification of a lower bound proof for OFL without predictions), the bound itself nicely complements the upper bound as well.

My main criticism of the paper is the presentation of the lower bound, in particular the claim that the lower bound shows that a bound on $\eta_1$ instead of $\eta_\infty$ is unlikely to offer substantial improvements. The authors state that their lower bound holds even when $\eta_1 = O(1)$. However, in the lower bound example, the distances in the metric could be smaller than one, i.e. the ratio $\eta_1/\eta_\infty$, which is e.g. not changed by just rescaling the metric, and is in my view the right quantity to look at, might still be quite large. So this doesn't say anything about the relation between $\eta_1$ and $\eta_\infty$, i.e. this ratio can vary depending on the setting of $\eta_\infty$ in their example (roughly, $\eta_1$ is fixed but $\eta_\infty$ can be varied). I think if $\eta_\infty$ is sufficiently small in their lower bound one gets $\eta_1 = O(\eta_\infty)$, which shows that in general no $o(\frac{\log n\eta_1/OPT}{\log \log n})$-competitive algorithm is possible, which would e.g. say that one can't get a better asymptotic ratio by replacing $\eta_\infty$ with $\eta_1$, but I'm not completely sure if this is the right takeaway, and the authors don't spell out any argument like this. I would like to see the relationship between $\eta_1$ and $\eta_\infty$ stated more clearly in this lower bound, as well as a discussion of what this relationship implies about bounds depending on $\eta_1$, if the authors wish to make claims that $\eta_1 \ll \eta_\infty$ is unlikely to help like they do in the introduction.

Also, I don't believe it is mentioned anywhere that OPT does not have to be the offline optimal solution in the analysis, which is implicitly being used in the lower bound - in this lower bound, $\eta_\infty$ is the distance between the predictions and a solution which may be optimal. For clarity's sake it would be good to either mention this somewhere, or the lower bound construction should prove the one-facility solution is optimal; if anything the former should strengthen the upper bound result by making it more general, and it would be more in line with the empirical results, which don't find an exact solution to base predictions off

Lastly, less so a weakness than a lack of strength, while online facility location is a fairly fundamental question in the online algorithms space, I don't think that compared to e.g. "The Primal-Dual method for Learning Augmented Algorithms" by Bamas et al., this paper's techniques are fundamentally breaking new technical ground in the learning-augmented space that could offer insights into other problems. Again, I wouldn't view this a weakness, just a reason why I don't feel I can rate this paper highly despite enjoying reading it.

**Summary Of The Paper:**

The paper considers the online facility location problem with predictions. In this problem, there is a set of facilities given offline on a metric and a sequence of demand points appearing online. When a demand point arrives, we must either route it to a facility we have previously opened, or open new facilities and route the demand to one of the new facilities. The total cost of a solution is the total cost of facilities opened, plus the total distance between each demand point and the facility it was routed to. In the learning-augmented setting, when each demand point arrives we are also given a prediction, in the form of a facility we believe it is routed to in an optimal solution. Let $\eta_\infty$ be the maximum distance between any the facility we predict a demand point is routed to in the optimal solution, and the facility it is actually routed to in the optimal solution. The authors give a $O(\log \frac{n \eta_\infty}{OPT})$-competitive algorithm, where $n$ is the number of demand points and $OPT$ is the cost of the optimal solution - without loss of generality, we can assume $\eta_\infty = O(OPT)$, so this is at worst an $O(\log n)$-competitive algorithm. They also show that this is the optimal competitive ratio up to $O(\log \log n)$ factors, even if $\eta_1$ (the total distance between predictions and OPT's facilities rather than the maximum) is constant. The authors also give experiments showing that on real-world data sets with synthetic predictions that have a bounded $\eta_\infty$ value, their algorithm outperforms Meyerson's when the prediction error is low, outperforms following the predictions when the prediction error is high, and is not too much worse than Meyerson's when the prediction error is high.

The algorithm the authors use first uses the algorithm of Meyerson for the setting without predictions. In this algorithm, for each power of 2 $2^k$, we consider opening the closest facility to the demand point whose cost is at most $2^k$, or that is already in our set of open facilities. Let $\delta_k$ be the distance from the demand point to this facility. We open this facility with, roughly speaking, probability proportional to $\frac{\delta_{k-1} - \delta_k}{2^k}$. Intuitively, we will never open a facility that is further from the demand point from a previously opened facility, and the probability we open a facility closer than the best previously opened facility is roughly proportional to the decrease in connection cost from opening this facility versus the best facility costing half as much, divided by this facility's opening cost. Following a round of Meyerson's algorithm, we use a "prediction step" to possibly open more facilities, whose expected cost is at most the increase in facility and connection cost in the round of Meyerson's algorithm. To do so, we consider facilities in a ball around the predicted facility whose radius is proportional to the distance from the predicted facility to the nearest facility opened in a previous prediction step, and consider opening the cheapest one. We will definitely open it if doing so would not cause the cost of the prediction step to exceed the cost of the round of Meyerson's algorithm, otherwise we open it with some probability, such that the prediction step's expected cost is exactly the cost of the round of Meyerson's algorithm.

**Summary Of The Review:**

Overall I think the paper is a weak accept, if the aforementioned concerns with the presentation of the lower bound construction can be resolved (if they are not resolved in the rebuttal period, I would rate at a weak reject). I think the algorithm and its analysis are fairly non-trivial but intuitive and enjoyable to read about, the problem being addressed is one of interest, and the lower bound/empirical results nicely complement the upper bounds. However, it is not clear to me if this paper stands out in particular amongst other papers in the learning-augmented online algorithms literature, and there are some concerns with the presentation of the lower bound I wish to see addressed before publication.

---

> ### Author Response · Authors · 2021-11-14
> **Response to m2EN**
>
> **Q1**: My main criticism of the paper is the presentation of the lower bound, in particular the claim that the lower bound shows that a bound on \eta_1 instead of \eta_\infty is unlikely to offer substantial improvements. ... this doesn't say anything about the relation between \eta_1 and \eta_\infty, i.e. this ratio can vary depending on the setting of \eta_\infty in their example (roughly, \eta_1is fixed but \eta_\infty can be varied). ... I would like to see the relationship between \eta_1 and \eta_\infty stated more clearly in this lower bound, as well as a discussion of what this relationship implies about bounds depending on \eta_1, if the authors wish to make claims that \eta_1 \ll \eta_\infty is unlikely to help like they do in the introduction.
>
> **A1**: We thank the reviewer for the very detailed comments!
>
> We rewrote the statement as well as the proof of the lower bound part in our revision. In our new statement, we first fix the opening cost to be 1 (which is w.l.o.g. since one can rescale the metric), then we show for every $\eta_\infty \in (0, 1]$, there is a class of inputs such that no online algorithm is $o(\log (n \eta_\infty / OPT) / \log \log n)$-competitive, and this holds even when $\eta_1 = O(1)$.
>
> So, in the parameter regime of this lower bound, $\eta_\infty$ is a constant, and $\eta_1 = O(1)$ is constant times larger than $\eta_\infty$, which rules out the ratio of $O(\log (\eta_1 / OPT) / \log \log n)$.
>
> **Q2**: Lastly, less so a weakness than a lack of strength, while online facility location is a fairly fundamental question in the online algorithms space, I don't think that compared to e.g. "The Primal-Dual method for Learning Augmented Algorithms" by Bamas et al., this paper's techniques are fundamentally breaking new technical ground in the learning-augmented space that could offer insights into other problems. Again, I wouldn't view this a weakness, just a reason why I don't feel I can rate this paper highly despite enjoying reading it.
>
> **A2**: We view this differently. Conceptually, OFL is inherently a geometric problem, and geometric problems seem to be very much less studied in the field of learning-augmented algorithms. The algorithmic techniques developed towards solving OFL with predictions, could be useful to solving other geometric problems in the learning-augmented setting, including but not limited to capacitated facility location and k-clustering problems.

---

> > ### Comment · Reviewer_m2EN · 2021-11-18
> > **Will re-read!**
> >
> > Thank you for your response, I am excited to see/read the revision. After briefly skimming, I believe my concerns with the lower bound's presentation have been adequately addressed by the revision. After reading more closely, I will update the initial review to reflect that the concerns have been addressed.

---

### Official Review · Reviewer_3sjS · 2021-11-05

**Correctness:** 4
**Technical Novelty And Significance:** 3
**Empirical Novelty And Significance:** 4
**Recommendation:** 6
**Confidence:** 4

**Main Review:**

Strengths:

1. The paper is technically solid, and the analysis, although it builds heavily on Meyerson's analysis, is non-trivial. I would say that from a purely technical perspective, the paper passes the bar for the conference.
2. The facility location problem is an important one. While I am generally inclined more toward papers that develop general purpose tools for the online algorithms with predictions area instead of giving very specific algorithms for particular problems, I think online facility location is a problem that is fundamental enough that it merits study in this framework. (As the paper mentions, this is not the only recent paper on this very problem, although the results are in this paper are more general in that they apply to on-uniform opening costs.)

Weaknesses:

1. I am really not convinced by the error model in this paper. It bases the error on the worst prediction, which means a large error even with one bad prediction (and every other prediction is precisely correct). The paper tries to justify this by showing that the worst case error cannot be replaced by the sum of prediction errors. But, to me, the right solution to this would be to have two components of the error: one component that counts the number of mispredictions (where the distance by which the prediction missed is irrelevant), and the other that captures the $\ell_1$ distance for the correct predictions (that might still not be exact). (Of course, what is counted as a misprediction and what is counted as a correct prediction is not canonical, but the algorithmic guarantee can hold for any such choice.) This would naturally interpolate between robustness (where all predictions are counted as mispredictions) and consistency (where all predictions are correct predictions and the error distance is 0). Instead, this paper replaces $\ell_1$ error with a very loose upper bound of $n$ times the $\ell_\infty$ error, which as I mentioned above, wouldn't give anything better than an online algorithm (without predictions) even if one prediction is badly off.

2. My second concern (and question to the authors) is the dependence of this algorithm on Meyerson's algorithm. The way the algorithm is set up, it takes the solution produced by Meyerson's algorithm and uses the budget provided by this algorithm to essentially reduce future costs by buying facilities "near" the predicted facility. What if we replaced Meyerson's algorithm with another competitive  facility location algorithm? If the algorithm can use any competitive facility location algorithm, that has multiple benefits: (a) the algorithmic framework being designed is more flexible, and can be applied to other similar problems instead of designing another algorithm from scratch, and (b) much of the analysis, which at the moment reprove properties of Meyerson's algorithm, can be eliminated and simplified. As stated, the current algorithm seems to rely on specific properties of Meyerson's algorithm, but is that really necessary?

3. One minor gripe is the way the algorithm is presented in this paper. Could you please include a text description (perhaps in addition to the pseudocode) so that one doesn't have to understand the algorithm solely from the pseudocode?

**Summary Of The Paper:**

This paper presents an algorithm for online facility location with predictions. The prediction model is that on the arrival of each client, the prediction provides a facility to connect this client to. The quality of the prediction is quantified in the result by the maximum prediction error, i.e., the maximum distance between the locations of the predicted facility and a facility serving the same client in a fixed optimal solution. The main result is that the competitive ratio is constant (which is the best offline ratio) when the maximum prediction error is small (e.g., when it is a 1/n-fraction of the optimal cost, which of course if true when the predictions are precisely correct), and O(log n) (which is the best online ratio, ignoring lower order terms), when the prediction error is large. The algorithm is an augmentation of an online algorithm for the problem by Meyerson that obtains a competitive ratio of O(log n), where in each step in addition to running Meyerson's algorithm, the algorithm uses a second copy of the incremental cost in Meyerson's step to preemptively open more facilities "close" to the predicted facility for the current client. In addition to the theoretical analysis, the paper includes an experimental evaluation of the algorithm comparing it to baselines produced by an algorithm that blindly trusts the prediction and Meyerson's algorithm.

**Summary Of The Review:**

This is a credible paper on online facility location with predictions. The problem considered is important and relevant, and the results obtained are tight in the sense that consistency and robustness results in the right ballpark. But, I have concerns about the error metric, which is the most important parameter here - with the current error metric, the very realistic possibility of having a single bad prediction is also going to relegate the guarantees of the algorithm to those of having no prediction at all. I would also have been happier with an algorithm that can use an online algorithm for the problem in a more black box manner - this would make the framework more generally applicable (and I suspect this is possible even with the current framework, or some modification of it, because I don't see the framework as conceptually using anything specific about Meyerson's algorithm, although I might be wrong on that last point).

---

> ### Author Response · Authors · 2021-11-14
> **Response to 3sjS**
>
> **Q1**: I am really not convinced by the error model in this paper. It bases the error on the worst prediction, which means a large error even with one bad prediction (and every other prediction is precisely correct). The paper tries to justify this by showing that the worst case error cannot be replaced by the sum of prediction errors. But, to me, the right solution to this would be to have two components of the error: one component that counts the number of mispredictions (where the distance by which the prediction missed is irrelevant), and the other that captures the $\ell_1$ distance for the correct predictions (that might still not be exact). (Of course, what is counted as a misprediction and what is counted as a correct prediction is not canonical, but the algorithmic guarantee can hold for any such choice.) This would naturally interpolate between robustness (where all predictions are counted as mispredictions) and consistency (where all predictions are correct predictions and the error distance is 0). Instead, this paper replaces $\ell_1$ error with a very loose upper bound of $n$  times the $\ell_\infty$ error, which as I mentioned above, wouldn't give anything better than an online algorithm (without predictions) even if one prediction is badly off.
>
> **A1**: We thank the reviewer for this insightful suggestion.
>
> We added a new section (Section F) to discuss this in the revised draft. In particular, we tried to formalize your idea by introducing an integer parameter $t$ ($1 \leq t \leq n$), and let $\eta_\infty^{(t)}$ be the $t$-th largest prediction error, and define $\eta_1^{(t)}$ similarly. That is, we look at the prediction errors ignoring $t - 1$ “outliers”, i.e., the $t - 1$ largest errors. Our key findings are summarized in the following.
>
> - Our algorithm, without any change (but with an improved analysis), actually has a ratio of $O(\log t+\log (n \eta_{\infty}^{(t)} / OPT))$ for **every** $t$, which is equivalent to $\min_{1 \leq t \leq n} O(\log t + \log (n \eta^{(t)}_\infty / OPT))$. We obtain this by slightly improving our analysis, using some simple observations.
>
> - Our lower bound still suffices to rule out the possibility of replacing $\eta^{(t)}_\infty$ with $\eta^{(t)}_1$ for the abovementioned bound. However, it is indeed an interesting question to explore whether or not a bound like $O(\log t + \log (\eta^{(t)}_1 / OPT))$ for some **specific** $t$, such as $t \geq \sqrt{n}$, could hold.
>
> We will try to incorporate this part into the main body of the paper in a later revision and acknowledge your suggestion.
>
> **Q2**: My second concern (and question to the authors) is the dependence of this algorithm on Meyerson's algorithm. The way the algorithm is set up, it takes the solution produced by Meyerson's algorithm and uses the budget provided by this algorithm to essentially reduce future costs by buying facilities "near" the predicted facility. What if we replaced Meyerson's algorithm with another competitive facility location algorithm? If the algorithm can use any competitive facility location algorithm, that has multiple benefits: (a) the algorithmic framework being designed is more flexible, and can be applied to other similar problems instead of designing another algorithm from scratch, and (b) much of the analysis, which at the moment reprove properties of Meyerson's algorithm, can be eliminated and simplified. As stated, the current algorithm seems to rely on specific properties of Meyerson's algorithm, but is that really necessary?
>
> **A2**: Unfortunately, our current analysis did use important properties of Meyerson’s algorithm, and replacing it with an arbitrary competitive algorithm seems not to work. We highlight two important properties that we use from Meyerson’s.
>
> 1. The behavior of the long-short distance stage. In particular, this could also be understood as an observation stated in our Section 1.2: Suppose before the first demand point arrives, the algorithm is already provided with an initial facility set $\bar{F}$, such that for every facility $f^*$ opened in $OPT$, $\bar{F}$ contains a facility $f’$ that is of distance $t$ to $f^*$. Then Meyerson’s algorithm is $O(\log (n t / OPT))$-competitive.
>
> 2. In the long-short transition round, the cost of the competitive algorithm has to be at most $O(\log (n \eta_\infty / OPT)$ times the optimal cost $w(f^*)+\sum_{i \in [m]} d(x_i, f^*)$; this is precisely the statement of Lemma 3.7 (and we state/show it is actually $O(1)$ for Meyerson’s). Note that this property cannot be obtained if one only assumes the algorithm is $\rho$-competitive, where $\rho$ has to be at least $\Omega(\log n / \log \log n)$ because of the known lower bound.
>
>
> **Q3**: Could you please include a text description so that one doesn't have to understand the algorithm solely from the pseudocode?
>
> **A3**:  In the revision, we added more explanations to the algorithm at the beginning of Section 3.

---

### Official Review · Reviewer_TTSB · 2021-11-07

**Correctness:** 4
**Technical Novelty And Significance:** 3
**Empirical Novelty And Significance:** 1
**Recommendation:** 6
**Confidence:** 5

**Main Review:**

Strengths of the paper:
The paper is reasonably well written for most parts -- except perhaps please provide a bit more description of the algorithm instead asking users parse through the pseudocode. The non-uniform case requires a non-trivial amount of analysis and may be of interest to the community.

Weaknesses of the paper:

While theoretically the problem is interesting -- I am not convinced of the practicality of the model. How will you construct a model that can predict what facilities an optimal solution would pick -- it depends on the other facilities that have already been chosen -- so what dataset would you use to train such a model?   In the "real datasets" based experiments, this important aspect is simulated using a 3-approximation algorithm to find the optimal solution and then randomized in accordance with "prediction error" n_\infty.  This needs to be clarified.
While the non-uniform case is interesting -- as opposed to the uniform case, which is down right trivial, and needs a good bit of work to prove, the idea is quite straightforward and predictable. Indeed, the doubling trick (used because, we do not need n_\infty) is quite common place in the field of approximation algorithm -- ski-rental problem (Purohit et al) for example uses a similar trick.
Lastly, as a pedantic issue, I would really question the suitability of this work in ICLR. There is no ML/AI, if one discounts the vague motivation (which I have issues with as well).


**Summary Of The Paper:**

The paper studies the classic problem of Online Facility Location augmented with a prediction model. The main contribution is a logarithmic competitive ratio algorithm for the case where facility opening costs are non-uniform.

**Summary Of The Review:**

The key issue that needs to be addressed is the motivation for assuming that new facility locations can be predicted using a machine learning model. I feel this is quite unrealistic and significantly diminishes the value of the paper. It would be great if the authors could address this issue in the rebuttal phase. Otherwise, I think theoretically the paper is worth publishing, albeit at a more theoretical computer science (such as ESA, or APPROX ).

---

> ### Author Response · Authors · 2021-11-14
> **Reponse to TTSB**
>
> **Q1**: please provide a bit more description of the algorithm instead asking users parse through the pseudocode.
>
> **A1**: In the revision, we added more explanations to the algorithm at the beginning of Section 3.
>
> **Q2**: I am not convinced of the practicality of the model. How will you construct a model that can predict what facilities an optimal solution would pick -- it depends on the other facilities that have already been chosen -- so what dataset would you use to train such a model?   In the "real datasets" based experiments, this important aspect is simulated using a 3-approximation algorithm to find the optimal solution and then randomized in accordance with "prediction error" n_\infty.
>
> **A2**: See our common response (https://openreview.net/forum?id=DSQHjibtgKR&noteId=ikEfhfyZ7bp) about an easily-obtainable predictor and the evaluation of its performance.
>
> **Q3**: While the non-uniform case is interesting -- as opposed to the uniform case, which is down right trivial, and needs a good bit of work to prove, the idea is quite straightforward and predictable. Indeed, the doubling trick (used because, we do not need n_\infty) is quite common place in the field of approximation algorithm -- ski-rental problem (Purohit et al) for example uses a similar trick.
>
> **A3**: We are not sure whether we understand this comment correctly, especially which part this “doubling trick” refers to, but it seems the reviewer is talking about line 18 - 19 of our algorithm, since these two lines have a behavior of shrinking the radius r by half, then finding a candidate open facility whose cost is 2 times larger.
>
> While this is an important step whose aim is to find a facility that is close enough to $\eta_\infty$ while balancing the opening cost, it is by no means the only important technical step in the proof, and other nontrivial ideas are also needed.
>
> For instance, a crucial technical insight is that our algorithm behaves in long-short distance stage.
>
> - In the long stage, our Prediction step helps the Meyerson step to make fast progress, so that it transits to short stage very quickly;
> - Then crucially, in the start of the short stage, a facility within distance $O(\eta_\infty)$ to the optimal facility is built, and we show once this happens, the Meyerson step itself already suffices to guarantee our desired ratio, while the Prediction step guarantees the opening cost is comparable to the cost made by the Meyerson step.
>
> **Q4**: Lastly, as a pedantic issue, I would really question the suitability of this work in ICLR. There is no ML/AI, if one discounts the vague motivation (which I have issues with as well).
>
> **A4**: We justified how our predictor can be found in the common response (https://openreview.net/forum?id=DSQHjibtgKR&noteId=ikEfhfyZ7bp), so we hope this helps to address the concern about the motivation of the paper.
>
> Regarding the impact in AI/ML, apart from our added discussion about the predictors, a conceptual contribution of our work is that it hints how a good predictor should behave when one would like to construct it in practice - for instance, we identify \eta_\infty as an important parameter that predictors should aim to minimize this.

---

### Official Review · Reviewer_tWdT · 2021-11-08

**Correctness:** 4
**Technical Novelty And Significance:** 3
**Empirical Novelty And Significance:** 2
**Recommendation:** 8
**Confidence:** 4

**Main Review:**

Overall, the paper is very well-written and does a good job of explaining the key ideas behind the algorithm and proof techniques. The algorithm itself is quite natural and also likely to be useful to practitioners - essentially, the proposed algorithm is to use Myerson's algorithm for online facility location and then open an additional facility at the predicted location whenever Myerson decides to open one. The key technical contributions are for the variant when facilities have non-uniform costs, in which case the additional facilities are opened "near" the predicted location.

The proposed algorithm obtains a competitive ratio of $O(log (n \eta_\infty/OPT))$ where $\eta_\infty$ is the maximum prediction error. So the algorithm only improves upon the worst-case setting as long as each prediction is $<< OPT$. The authors also demonstrate a matching lower bound though, and in particular show that the dependence on $n \eta_\infty$ is unavoidable.

Q. Is it easy to compute the "consistency" of the algorithm, i.e., the competitive ratio when the predictions are perfectly accurate? Can one try to obtain a $(1+\epsilon)$-consistent (but also robust) algorithm?
Q. Can the robustness bound be improved to $O(\log n / \log \log n)$?

Minor comments:
- Abstract: The last sentence of the first paragraph needs some punctuation
- Page 4: Last line of Section2. "0\leq k \leq L" --> "1 \leq k \leq L" ?
- Page 5. Last line. "in by Lemma 3.5" --> "is by ..."
- The statement of Lemma 3.6 bounds $cost_M()$, but the proof only bounds $cost_P()$. It will be good to remind readers that the expected cost is the same.





**Summary Of The Paper:**

The paper studies a variant of the online facility location problem where each demand point arrives with a prediction of which facility it is assigned to in an optimal solution. In line with recent work on learning-augmented algorithms, the paper designs an algorithm whose performance degrades gracefully with the quality of the predictions and yet retains an almost optimal worst-case guarantee.

**Summary Of The Review:**

Overall the paper is a nice addition to the new area of learning-augmented algorithms. The proposed algorithm is practical and almost optimal, though it would be nice to obtain the optimal robustness bound of $O(\log n / \log \log n)$. The paper seems to be a good fit for ICLR.

---

> ### Author Response · Authors · 2021-11-14
> **Response to tWdT**
>
> In the following, we respond to the specific questions/concerns raised by the reviewer. The minor comments have been addressed in the revised draft.
>
> **Q1**: Is it easy to compute the "consistency" of the algorithm, i.e., the competitive ratio when the predictions are perfectly accurate?
>
> **A1**: The competitive ratio in this case is $O(1)$. We added an explicit statement in Theorem 1.1 & 3.1 to clarify this, and the proof has also been changed to reflect this in our revision.
>
> **Q2**: Can one try to obtain a (1+eps)-consistent (but also robust) algorithm?
>
> **A2**: This is an interesting question, but we suspect such an algorithm does not exist. Our intuition is that, in order to be $(1 + \epsilon)$-approx, the algorithm has to follow the prediction very closely. However, when the prediction is inaccurate by a very slight factor, following the prediction could incur a huge error. Hence, the algorithm has to be able to somewhat distinguish the slightly inaccurate prediction from the perfectly accurate one, but this seems to be very difficult.
>
> **Q3**: Can the robustness bound be improved to $O(\log n / \log \log n)$?
>
> **A3**: Since our algorithm is based on Meyerson’s algorithm, it would be a first step to figure out whether or not the original Meyerson’s algorithm is $O(\log n / \log \log n)$-competitive or not. Unfortunately, although it was mentioned in [Fotakis, Algorithmica 2008] that improved analysis of Meyerson’s could lead to $O(\log n / \log \log n)$ ratio, we did not see a complete formal proof of this, especially for the non-uniform opening cost case.
>
> Hence, although there is evidence that one can improve to $O(\log n / \log \log n)$ for robustness, we have not figured out the required technicalities for the time being.

---

### Official Review · Reviewer_m1RY · 2021-11-08

**Correctness:** 4
**Technical Novelty And Significance:** 3
**Empirical Novelty And Significance:** 2
**Recommendation:** 6
**Confidence:** 4

**Main Review:**

Strength of this paper: The paper gives a natural relation between prediction error and competitive ratio for a classical online problem. Furthermore, the lower bound presented in the paper makes the competitive ratio of their algorithm near-optimal. Due to this, I find overall result in this paper to be a nice one.

On the technical side, the authors present a fine-grained analysis of Meyerson’s algorithm parameterizing the competitive ratio on the prediction error. Although it appears somewhat straight-forward, it seems to be a nice contribution.

Weakness:

I have three sets of concerns/criticisms.

a)	Their algorithm uses the worst-case algorithm by Meyerson as a black box to guide its decisions. A more natural approach would have been to incorporate predictions within Meyerson’s algorithm. Their experiments seem to suggest that even with perfect predictions, the empirical gains over Meyerson’s algorithm seem only minor. I suspect that relying on Meyerson’s cost as a black box may be the main reason for this. Indeed, the authors explicitly state that they do not make any attempt to reduce the hidden constant within the big-O notation of the competitive ratio. So, there may be other algorithms that have a similar theoretical trade-off but do significantly better in practice.

b)	There are several presentation issues in the paper. These include undefined notations, typos, and (minor) inaccurate lemma/theorem statements. I point some of them out below. These issues make it challenging to fully verify and appreciate the theoretical claims, especially in a short time frame. Experimental result section is somewhat confusing as well. The authors do not explain their choice of data sets and what makes these data sets “real” for the facility location problem. Also, how are the non-uniform costs assigned to facilities for the non-uniform data set?

c)	Finally, I do not see why this prediction model is natural for this problem. How can we use historical data to generate predictions with small \eta value?


Presentation issues are listed below:

1)	OPT is defined to be a value sometimes and a set of facilities on some other occasions (For instance in the statement of theorem 1.1)
2)	Can $\eta_\infty$ be 0? If so, how should we interpret your competitive ratio?
3)	Section 1.2 paragraph 2, there is a missing \log is the competitive ratio of Meyerson’s algorithm
4)	Section 1.3 determined -> deterministic
5)	$f_{open}$ is not initialized in the procedure PRED. L23 of the algorithm uses it without any initialization
6)	A short overview of Meyerson’s algorithm in the Preliminaries can be very helpful.
7)	Lemma 3.5 $X(f^*)$ is undefined.
8)	Theorem 1.2 Should it be for $\eta_{\infty} > 0$?
9)	Page 6: “we does” does not seem correct.
10)	Page 6, line 1 “that” -> “such that”. This seems to be a recurring issue.
11)	There are several other typos. A careful proofreading of the paper is necessary for it to be of publishable quality.


**Summary Of The Paper:**

This paper studies the classical online facility location problem in a metric space. Given a new demand in this space, the algorithm will either open a new facility by paying an opening cost $w(f)$ and assign the demand node to this facility, or it will assign the demand node to an existing open facility. The cost incurred by the algorithm is the sum of all the opening costs and assignment costs.

In this paper, the authors consider the setting where, along with every demand node, we are also given a prediction on which facility it gets assigned to in the optimal. Such predictions can potentially be acquired through historical data.

Their main contribution is an online algorithm for facility location with predictions that has a near-optimal competitive ratio. The competitive ratio relies on a parameter $\eta_\infty$ which is simply the largest prediction error, where prediction error for any demand point is the distance between the predicted facility and the facility that is assigned to the demand in the optimal solution.
Intuitively, the authors show that if the predictions are accurate (i.e., $\eta_\infty$ is small), then their algorithm is $O(1)$-competitive. On the other hand, if the predictions are highly inaccurate, the algorithm will become $O(\log n)$ competitive.

The algorithm relies on an $O(\log n)$ competitive worst-case optimal (randomized) online algorithm by Meyerson. The idea becomes very simple when facilities have uniform opening costs: Suppose Meyerson’s algorithm opens a facility, one can simply also open a facility at the predicted location. By doing so, the cost incurred will not be more than twice that of Meyerson’s algorithm which is $O(\log n)$-competitive. However, when the predictions are accurate, opening the facilities at predicted locations lead to better future assignments leading to $O(1)$-competitive algorithm.

They extend this approach in a non-trivial way to the case with non-uniform opening costs.


**Summary Of The Review:**

Due to the weaknesses mentioned above, I do not believe the paper is ready for publication.

Update: Thank you for addressing my concerns. After reading the new version, I am happy to update my score.

---

> ### Author Response · Authors · 2021-11-14
> **Response to m1RY**
>
> **Q1**: Their experiments seem to suggest that even with perfect predictions, the empirical gains over Meyerson’s algorithm seem only minor. I suspect that relying on Meyerson’s cost as a black box may be the main reason for this.
>
> **A1**:  In fact, the use of Meyerson’s algorithm is **not** in a black-box way. In particular, even though we run Meyerson’s as the first step in every iteration of the algorithm, the second, Prediction step, could greatly affect the behavior of subsequent Meyerson steps. This forces us to open up this black box, particularly in the analysis.
>
> We also emphasize that our Prediction step is tailored to Meyerson’s, and replacing the Meyerson step with another algorithm may not work. In particular, we need to use the following property of Meyerson’s.
>
> 1. The behavior of the long-short distance stage. In particular, this could also be understood as an observation stated in our Section 1.2: Suppose before the first demand point arrives, the algorithm is already provided with an initial facility set $\bar{F}$, such that for every facility $f^*$ opened in $OPT$, $\bar{F}$ contains a facility $f’$ that is of distance $t$ to $f^*$. Then Meyerson’s algorithm is $O(\log (n t / OPT))$-competitive.
>
> 2. In the long-short transition round, the cost of the competitive algorithm has to be at most $O(\log (n \eta_\infty / OPT)$ times the optimal cost $w(f*)+\sum_{i \in [m]} d(x_i, f^*)$; this is precisely the statement of Lemma 3.7 (and we state/show it is actually $O(1)$ for Meyerson’s). Note that this property cannot be obtained if one only assumes the algorithm is $\rho$-competitive, where $\rho$ has to be at least $\Omega(\log n / \log \log n)$ because of the known lower bound.
>
> This is our technical reason to use Meyerson’s algorithm, and this should also dispel the suspicion that we use Meyerson’s algorithm mainly because of its good practical performance.
>
> **Q2**: Indeed, the authors explicitly state that they do not make any attempt to reduce the hidden constant within the big-O notation of the competitive ratio. So, there may be other algorithms that have a similar theoretical trade-off but do significantly better in practice.
>
> **A2**: Usually, reducing the constant hidden in the big-O is more about tightening the analysis, and the (loose) hidden constants do not necessarily translate to the (bad) practical performance of the algorithm. Actually, even the worst-case ratio (which is the not-hidden part in the big-O) is often not directly related to the practical performance as well. For instance, as can be seen from our experiments, even though Meyerson’s algorithm has a ratio of $O(\log n)$, its empirical performance seems much better than that. Moreover, our algorithm, even provided with relatively inaccurate predictions, outperforms Meyerson’s algorithm. Therefore, we do not see an absolute necessity to tighten the hidden constant in the analysis, or find an algorithm that excels in this regard.
>
> However, we do agree that there might be other algorithms that have better practical performance. But we also observe that significant improvement over our algorithm seems unlikely on our tested datasets, since even Meyerson’s algorithm (without predictions) has a close-to-one empirical ratio.
>
> **Q3**: The authors do not explain their choice of data sets and what makes these data sets “real” for the facility location problem. Also, how are the non-uniform costs assigned to facilities for the non-uniform data set?
>
> **A3**: These datasets have been used in various previous works that study clustering/facility location problems. For instance,
> - Twitter dataset was used in “Fully Dynamic Consistent Facility Location, Cohen-Addad et al, NuerIPS 19” and “Fully Dynamic k-Center Clustering, Chan et al, WWW 18”
> - Adult dataset was used in “Fair clustering through fairlets, Chierichetti et al, NeurIPS 17”
>
> Moreover, US-PG is a power grid dataset, where power nodes may be viewed as facilities and/or demands, so the facility location problem is defined naturally.
>
> The Non-Uni dataset is a road network dataset extracted from OpenStreetMap, where each point corresponds to a geographical object (e.g., a building), and the opening cost is the density of population around the point. This way of specifying opening cost is very intuitive, since the price of building a facility at a building is likely to be related to the population density.
>
> We also added a short version of this explanation to our draft.
>
> **Q4**: How can we use historical data to generate predictions with small $\eta$ value?
>
> **A4**: See our common response (https://openreview.net/forum?id=DSQHjibtgKR&noteId=ikEfhfyZ7bp) about an easily-obtainable predictor and the evaluation of its performance.
>
> **Q5**: There are several presentation issues in the paper.
>
> **A5**: Thank you very much for the detailed comments. We went through the paper carefully and fixed many presentation issues, especially those you mentioned.

---

### Official Review · Reviewer_DWJH · 2021-11-08

**Correctness:** 4
**Technical Novelty And Significance:** 3
**Empirical Novelty And Significance:** 3
**Recommendation:** 8
**Confidence:** 4

**Main Review:**

This paper follows a recent line of work in considering online algorithms with error-prone predictions.  The main results are an online algorithm for online facility location in this setting and a nearly matching lower bound, which are interesting.  The experimental evaluation is reasonable, but not overly exciting since the improvement over Meyerson's algorithm seem to be very narrow in some cases.  The presentation is mostly clear, but some improvements could be made, especially to the analysis.  Overall, I would be okay with accepting this paper.  See below for some further comments/suggestions.

This paper considers the predictions as coming from an oracle.  Recent work has started to explore the question of how to construct predictions for online algorithms from past data, e.g. [1,2,3] below.  It would be interesting to explore how to construct predictions for online facility location from past data, especially for an empirical evaluation.

[1] - Customizing ML Predictions for Online Algorithms.  Anand et al. ICML 2020

[2] - Learning Online Algorithms with Distributional Advice.  Diakonikolas et al. ICML 2021

[3] - Learnable and Instance Robust Predictions for Online Matching, Flows, and Load Balancing.  Lavastida et al. ESA 2021

Some suggestions for the related work section, which could be fleshed out more:
 - Purohit et al. 2018 also considers non-clairvoyant scheduling on a single machine
 - Secretaries with Advice.  Dütting, et al. EC 2021. also considers the secretary problem with predictions.
 - Flow Time Scheduling with Uncertain Processing Time. Azar et al. STOC 2021 and Online Scheduling via Learned Weights. Lattanzi et al. SODA 2020 consider flow time scheduling and online load balancing, respectively, in settings with error-prone predictions.

Minor comments/suggestions
 - At the end of page 1, "such as experts' advices" -> "such as expert advice"
 - In the "Calibrating predictions" paragraph, "second last" -> "second to last"
 - In the "Calibrating predictions" paragraph, $\frac{n \eta_\infty}{OPT} = O(\log n)$ should probably be $\frac{n \eta_\infty}{OPT} = O(n)$
 - In the proof of lemma 3.6, "the fact that we does not" -> "the fact that we do not"
 - Paragraph before lemma 3.8, "let $\bar{\ell}$ be the integer that" -> "let $\bar{\ell}$ be the integer such that"
 - In the experiments, the four plots seem to have different scales on the x-axis.  This seems to be due to the differing distance scales across each dataset.  It might be helpful for the presentation to either explain this in the text or maybe present the plots with something like $n\eta_\infty / OPT$ on the x-axis.

------------------------------

Edit after reading author responses:

The author's responses have cleared up my questions.  I appreciate the experiments with the simple predictor, which seems reasonable to implement in practice and performs well on the considered datasets.  I have raised my score to accept.


**Summary Of The Paper:**

This paper considers the online facility location problem with predictions.  In this problem, there is a sequence of points which arrive online and the algorithm must either open a facility to serve each point, paying the facility opening cost plus the distance to the opened facility, or it can assign the point to an already open facility, paying the distance only.  Additionally, each arriving point is given a prediction of a facility which serves it in an optimal solution. The goal is to be competitive with the offline optimal solution, and the competitiveness is parameterized by the error in the predictions.  Here, the prediction error is the maximum distance between a predicted facility and the facility in the optimal solution which serves a point.

The main result is an algorithm which is $O(\min (\log n, \log \frac{n\eta_{\infty}}{OPT}))$-competitive, where $\eta_\infty$ is the max error and $OPT$ is the optimal cost.  This improves over worst case lower bounds when $\log \frac{n\eta_{\infty}}{OPT} = o(\frac{\log n}{\log\log n})$, and nearly matches worst case bounds otherwise.  At a high level the algorithm first follows Meyerson's algorithm, then opens some additional facilities by taking into account the predictions.  In order to get the stated bound, the predictions are "calibrated" so that $\eta_\infty = O(OPT)$ so that $\log \frac{n\eta_{\infty}}{OPT} = O(\log n)$, thus it suffices to only prove the bound of $O(\log \frac{n\eta_{\infty}}{OPT})$.  A lower bound is also given, showing that the result is nearly tight.

Additionally, an experimental validation is carried out, comparing the proposed algorithm with Meyerson's algorithm and a naive algorithm which follows the predictions.  The experiments are performed on datasets where the underlying metric is either Euclidean or a graph metric.  The predictions are simulated to have a given level of error.  While the proposed method underperforms against blindly following the predictions when the error is small, it is more robust to large prediction errors and often comes close to Meyerson's algorithm when the prediction error is large.

**Summary Of The Review:**

This paper considers a fundamental online problem in the presence of predictions with a nearly tight result for this setting.  The experimental validation is reasonable.  Some improvements could be made to the presentation, but it is currently clear.

---

> ### Author Response · Authors · 2021-11-14
> **Response to DWJH**
>
> **Q1**: "The experimental evaluation is reasonable, but not overly exciting since the improvement over Meyerson's algorithm seem to be very narrow in some cases."
>
> **A1**: While this is true in our experiments on uniform opening cost datasets, we note that Meyerson’s algorithm (without predictions) already has a ratio very close to 1 in those cases, so the room for improving the ratio by using predictions is inherently very small.
>
> **Q2**: This paper considers the predictions as coming from an oracle. Recent work has started to explore the question of how to construct predictions for online algorithms from past data, e.g. [1,2,3] below. It would be interesting to explore how to construct predictions for online facility location from past data, especially for an empirical evaluation.
>
> **A2**: See our common response (https://openreview.net/forum?id=DSQHjibtgKR&noteId=ikEfhfyZ7bp) about an easily-obtainable predictor and the evaluation of its performance.
>
> **Finally**, we added the suggested references and fixed the typos in our revision.

---

### Author Response · Authors · 2021-11-14
**Thanks for the comments & response to a common concern**

We thank all reviewers for the insightful and very detailed reviews!

Since several reviewers raise a common concern about whether or not the assumed predictor can be constructed in practice, we give our response in this common thread, and we address reviewers’ other concerns in separate responses.

We added a new experiment that evaluates the performance of an easily-obtainable predictor in our revised draft (Section 4). The simple predictor merely uses a local (near-)optimal solution, on the training set plus the points seen so far, to generate predictions, and we observe that such a simple predictor already outperforms Meyerson’s baseline, which is especially significant in the non-uniform opening cost dataset. We expect more sophisticated predictors that make use of ML techniques and/or take advantage of signals/features in the dataset to achieve even better results.

Next, we provide a brief description of the predictor and how it is trained, and the details could be found at the end of Section 4 in our updated draft.

Initially, the predictor runs the 3-approximate MP algorithm on the training set. Then when the online algorithm is run on the test set, the predictor periodically reruns the MP algorithm, on the union of the training set and the already-seen test data. The prediction is simply the nearest facility in the 3-approximate solution maintained by the predictor.

To evaluate this predictor,  we take a random sample of 30% of the dataset as the training set, and the remaining 70% serves as the testing set. This random-sampling way of constructing training data implicitly captures the correlation of points between training set and test set.

---

### Decision · Program_Chairs · 2022-01-20

**Decision:**

Accept (Poster)

**Comment:**

This paper considers the recent line of work on algorithms with predictions.  They give new results on the online facility location problem. Overall, the reviewers felt the topic was of interest to the community.  There were some concerns about the error metric used and the overall framework. However, the majority of reviewers still felt the paper was interesting and I think the paper can be accepted.